# How J-chain ensures the assembly of immunoglobulin IgM pentamers

Chiara Giannone [ID][1,4][✉], Xenia Mess[2], Ruiming He[2], Maria Rita Chelazzi[1], Annika Mayer[2], Anush Bakunts [ID][1], Tuan Nguyen [ID][2], Yevheniia Bushman [ID][2], Andrea Orsi[1], Benedikt Gansen[1,2], Massimo Degano [ID][3], Johannes Buchner [ID][2][✉] & Roberto Sitia [ID][1][✉]

## Abstract

**Polymeric IgM immunoglobulins have high avidity for antigen and complement, and dominate primary antibody responses. They are produced either as assemblies of six µ2L2 subunits (i.e., hexamers), or as pentamers of two µ2L2 subunits and an additional protein termed J-chain (JC), which allows transcytosis across epithelia. The molecular mechanism of IgM assembly with the desired stoichiometry remained unknown. Here, we show in vitro and in cellula that JC outcompetes the sixth IgM subunit during assembly. Before insertion into IgM, JC exists as an ensemble of largely unstructured, protease-sensitive species with heterogeneous, non-native disulfide bonds. The J-chain interacts with the hydrophobic β-sheets selectively exposed by nascent pentamers. Completion of an amyloid-like core triggers JC folding and drives disulfide rearrangements that covalently stabilize JC-containing pentamers. In cells, the quality control factor ERp44 surveys IgM assembly and prevents the secretion of aberrant conformers. This mechanism allows the efficient production of high-avidity IgM for systemic or mucosal immunity.**

**Keywords** Antibody Biogenesis; ERp44; Mucosal Immunity; Non-Native Disulfides; Protein Quality Control
**Subject Categories** Immunology; Structural Biology

## Introduction

Immunoglobulin M (IgM) are the first line of defense in antibody-mediated immune responses (Boes, 2000). Their polymeric structure guarantees high avidity for antigens, compensating for the generally low affinity of their combining sites (Keyt et al, 2020). Two IgM heavy (µ) and two light (L) chains assemble to form $µ_2L_2$ 'monomeric' subunits (Wiersma and Shulman, 1995), which associate with each other to form $[µ_2L_2]_6$ 'hexamers' (Fig. 1A).

Ig-secreting cells produce an additional protein, the joining chain (JC) which is inserted into $[µ_2L_2]_5$JC "pentamers" (Chapuis and Koshland, 1974; Mihaesco et al, 1976) leading to IgM with different structural properties (Fig. 1A). JC binds to the penultimate residue (C575) of a conserved 18-amino acid tailpiece (tp) present at the C terminus of secretory µ-chains (Pasalic et al, 2017). C575 also forms the disulfide bonds that link flanking $µ_2L_2$ subunits. Structural studies revealed that the µtps from the five $µ_2L_2$ subunits form a β-sandwich core structure in IgM pentamers (Li et al, 2020b; Kumar et al, 2020). Two of the eight highly conserved cysteines of the JC, C15 and C69, form disulfide bonds with C575 in the two flanking µ subunits (Li et al, 2020b; Kumar et al, 2020; Johansen et al, 2000, 2001) (Fig. 1B and Appendix Fig. S1). The endoplasmic reticulum (ER) chaperones BiP and ERp44 play sequential roles in IgM quality control, patrolling µL assembly and polymerization, respectively (Feige and Hendershot, 2011; Anelli et al, 2007; Anelli and Sitia, 2008). While BiP accumulates in the ER, ERp44 cycles between the Golgi and ER (Tempio et al, 2021), preventing the secretion of proteins with reactive thiols and/or non-native disulfide bonds (Yang et al, 2016; Watanabe et al, 2017; Giannone et al, 2022), including unpolymerized IgM subunits (Pasalic et al, 2017; Giannone et al, 2022; Alberini et al, 1990; Sitia et al, 1990; Fra et al, 1993; Anelli et al, 2003).

The decision to insert a JC rather than a sixth subunit is key in targeting IgM toward mucosal or serum immunity (Johansen et al, 1999; Li et al, 2020a). Through JC, IgM pentamers bind polymeric Ig receptors (pIgR) and are transcytosed across epithelial cells (Brandtzaeg and Prydz, 1984). Thus, the formation of polymers with a 5:1 stoichiometry ensures transcytosis with minimal loss of avidity for antigen and complement. Despite its importance, however, little is known so far about the mechanism of JC folding and assembly into IgM. Here, combining in cellula and in vitro approaches, we unveil the mechanism that allows the specific integration of exactly one JC into IgM. We show that JC is a largely unstructured polypeptide rich in non-native disulfides. It covalently binds to C575 in the µtp only when they are part of $[µ_2L_2]_5$ pentamers. The encounter complex is formed by hydrophobic interactions between the JC and µtps at the core of IgM pentamer. These interactions guarantee the proper assembly of JC into IgM,

[1]Division of Genetics and Cell Biology. Università Vita-Salute San Raffaele and IRCCS Ospedale San Raffaele, Via Olgettina 58, Milan, IT, Italy. [2]Technical University Munich, School of Natural Science, Department of Bioscience, Center for Protein Assemblies, Ernst-Otto-Fischer-Strasse 8, 85748 Garching, Germany. [3]Division of Immunology and Infectious Diseases. Università Vita-Salute San Raffaele and IRCCS Ospedale San Raffaele, Via Olgettina 58, Milan, IT, Italy. [4]Present address: Department of Chemistry, Massachusetts Institute of Technology, Cambridge, USA. ✉E-mail: giannone@mit.edu; johannes.buchner@tum.de; sitia.roberto@hsr.it

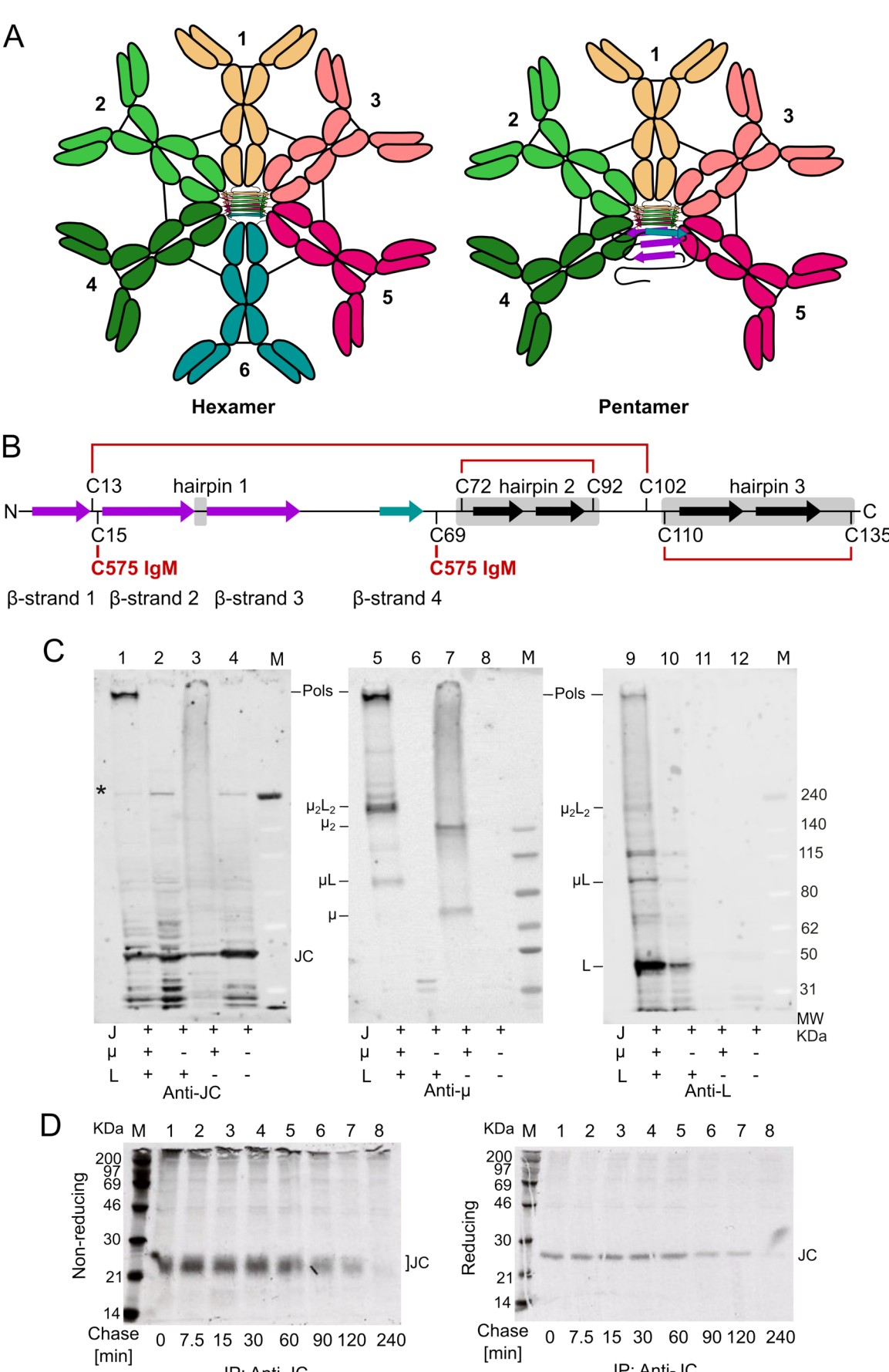

**Figure 1. Selective association of JC with nascent polymers.**

(A) Two types of IgM polymers. IgM are secreted in two main forms, hexamers and pentamers. Both share five $\mu_2L_2$ subunits (1–5) that form a β-sheet like core formed by the conserved C-terminal μtps and are stabilized by disulfide bonds linking C575 in adjacent subunits. The sixth element completing a planar hexamer can be a JC or another $\mu_2L_2$ subunit. (B) JC are rich in cysteines. Schematic representation of JC and its eight highly conserved cysteines, cysteine bridges (red lines), β-sheets (arrows) and hairpins (gray boxes) that are described in this paper. β-sheets are colored complementary to Fig. 1A. (C) JC associate only to IgM polymers. Aliquots of the lysates from J558L or NS0 myeloma transfectants expressing JC, L, and/or secretory μ chains as indicated were resolved under nonreducing conditions, and blots decorated sequentially with anti-JC, anti-μ and anti-L antibodies. Arrows point at the different assembly intermediates (μ, μL, etc.). Cells producing all three subunits secrete NP-specific IgM polymers, whilst those lacking L retain and degrade both μ and JC (Sitia et al, 1990; Fagioli and Sitia, 2001; Fagioli et al, 2001). The nature of the high molecular weight band decorated by anti-JC and indicated by the asterisk, is unknown. The experiment was repeated twice. (D) Unassembled JC are rapidly degraded. N[μ1] myeloma transfectants expressing JC and μ were pulsed for 5 min with radioactive aminoacids and chased for the indicated times before immunoprecipitation with anti-JC antibodies and protein A Sepharose beads (Fagioli and Sitia, 2001). The broad and heterogeneous signals of radioactive JC observed under nonreducing conditions (left panel) collapse in a sharp band upon reduction and alkylation (right panel). In lane 1 of the top panel, DTT present in the vicinal lane containing molecular weight markers reduced partly intra-chain disulfides of JC, slowing their gel mobility. The experiment was performed once. Source data are available online for this figure.

under the strict quality control of ERp44, and the secretion of polymers with the appropriate stoichiometry.

## Results

### JC binds to nascent IgM pentamers

We first analyzed JC insertion in IgM in murine myeloma transfectants that express it alone or with Ig λ light (L) and/or secretory μ chains (Fig. 1C). In J[μ1]L cells (Sitia et al, 1987, 1990), IgM secreting transfectants of the J558L myeloma that express B1-8-derived μ chain as well as endogenous L and J chains, μ and L chains efficiently assembled into intermediate complexes (μL and $\mu_2L_2$) and secretion-competent polymers recognized by anti-μ and anti-L antibodies (lanes 5 and 9, Pols). Only the latter were decorated by anti-JC (lane 1). In an NS0 myeloma transfectant expressing μ chain as well as endogenous JC, anti-μ antibodies decorated two sharp bands corresponding to μ and $\mu_2$ dimers, and a smear of higher molecular mass bands (lane 7). Also, anti-JC reacted with this smear (lane 3), suggesting that it likely consisted of heterogeneous, covalent assemblies of JC and μ chains (and possibly other molecules). The smear might reflect attempts to form stable polymers, doomed to fail in the absence of L chains (Hendershot et al, 1987; Sitia et al, 1990). In all cell types, including J558 cells (Oi et al, 1983) expressing J and L (lane 2) and NS0 (Galfrè and Milstein, 1981) cells expressing only JC (lane 4), anti-JC stained monomeric JC, and IgM polymers in J[μ1]L.

Pulse-chase experiments with $^{35}$S-labeled aminoacids showed that N[μ1] cells lacking L degrade JC after an initial lag (Fig. 1D), confirming previous results (Fagioli and Sitia, 2001). Interestingly, at all time points tested, monomeric JC migrated with a more diffuse pattern in nonreducing conditions (left panel) than in reducing conditions (right panel). JC collapsed into a sharp band upon DTT reduction, suggesting that JC may adopt multiple redox isoforms through the different arrangements of its intra-chain disulfide bridges, the nature of which is of particular interest in this study.

### JC outcompetes the sixth IgM subunit during assembly in cellula

To simplify the experimental system and bypass the requirement for L chain assembly, we exploited a Halo-tagged truncated μs variant lacking the variable and first constant domains (H-Cμ234tp) (Giannone et al, 2022). This construct and a variant lacking C575 were expressed in HEK293T cells in the presence or absence of JC (Fig. 2A). Cell lysates were resolved by nonreducing SDS–PAGE, and blots were first stained for the Halo ligand and then decorated with anti-JC antibodies to identify the components of the various assemblies. H-Cμ234tp chains polymerized efficiently in HEK293T, yielding a ladder with growing numbers of subunits, from H-Cμ234tp$_2$ 'monomers' to (H-Cμ234tp$_2$)$_6$ "hexamers". The slowest migrating band, consisting of secretion-competent hexamers (lane 1, indicated as 6X), was no longer detectable in cells co-expressing JC (lane 2, left panel). Anti-J antibodies decorated mainly two high molecular assemblies (lane 2, right panel). Based on its size, immunoreactivity, and fate, the most intense of the two likely consists of mature, secretion-competent JC-containing pentamers (dark blue asterisk in panel A). Noteworthy, lower order H-Cμ234tp assembly intermediates contained only traces of JC (light blue asterisks), indicating that also in non-lymphoid cells, JC forms stable covalent bonds mainly with pentamers. As expected, JC did not bind a H-Cμ234tp mutant lacking C575 (Fig. 2A, lanes 3-4, right panel). Very similar results were obtained with myc-tagged JC (Fig. EV1A).

Taken together, the above results suggested that JC covalently binds to nascent pentamers, outcompeting a sixth subunit. Alternatively, JC could bind to intermediates of IgM, subsequently recruiting other subunits to complete a hexameric structure. If this were the case, we would observe a change in the assembly pattern of IgM over the course of polymerization when JC is present compared to when it is not. This could eventually lead to the formation of covalent intermediates containing JC over time. To address this issue, we performed time-course experiments exploiting the covalent binding of the Halo-tag to its ligands (Tempio et al, 2021). Cells expressing H-Cμ234tp with or without JC were incubated with excess green R110 ligand to label preexisting molecules, washed, and cultured in the presence of red TMR ligand to follow the fate of newly made proteins (see scheme in Fig. EV1B). As expected, molecules present at time 0 were only green (compare Figs. 2B and EV1C): newly made, TMR-labeled H-Cμ234tp chains increased with time (Fig. 2B). Confirming our previous data, JC prevented the formation of hexamers, was mainly detected in high molecular weight polymers and bound H-Cμ234tp assembly intermediates very poorly. Clearly, JC had minor, if any, effects on the formation of dimers of H-Cμ234tp dimers or other intermediates, confirming that JC binds preferentially to nascent

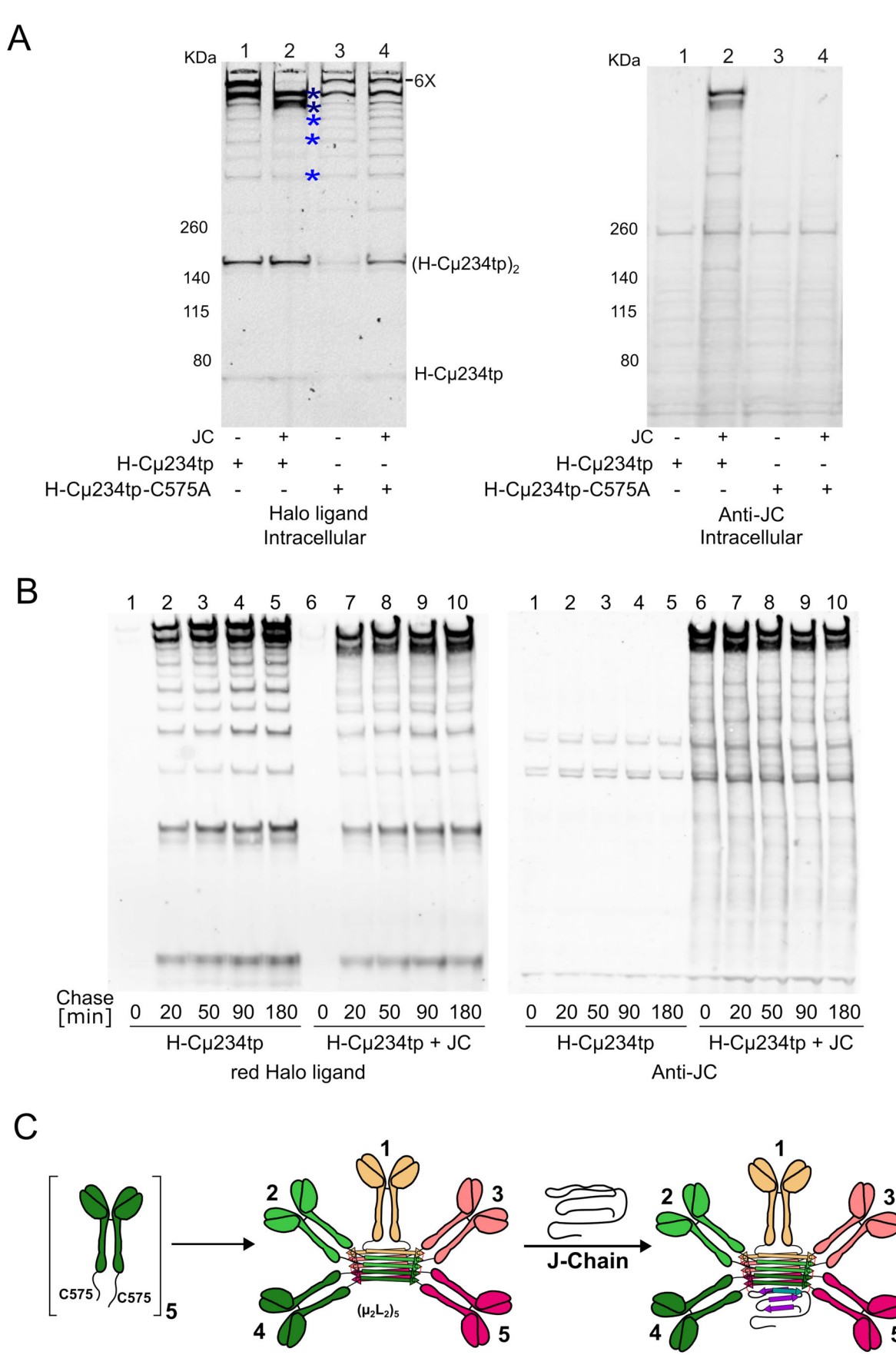

**Figure 2.   Assembly of pentameric IgM.**

(A) JC outcompete the sixth $\mu_2L_2$ subunit in associating with nascent pentamers. HEK293 cells expressing H-Cμ234tp chains with a cysteine (WT) or alanine in the penultimate position (C575A) were co-transfected with JC or empty plasmid, as indicated. Aliquots of their lysates were resolved under nonreducing conditions, and blots were decorated with Halo ligands (left panel) or anti-JC (right panel). In the absence of JC, two main assemblies accumulate intracellularly (lane 1). The slower band consists of secretion-competent (H-Cμ234tp$_2$)$_6$ hexamers (6X) and is the only species that negotiates secretion (see Figs. 6B and EV4A). Clearly, the presence of JC prevents the formation of hexamers and leads to the formation of low molecular weight polymers (dark blue asterisks), likely secretion-competent pentamers, and an uncharacterized form that is retained intracellularly (light blue asterisk). Note that also in HEK293 transfectants, JC interacts very weakly with H-Cμ234tp$_2$ and other low molecular weight intermediates (light blue asterisks in lane 2). The experiment was repeated at least three times with similar results. (B) Kinetics of IgM assembly. HEK293T cells were transiently transfected to express H-Cμ234tp or H-Cμ234tp + JC. Forty-eight hours after transfections, cells were treated as specified in Fig. EV1B to conduct a Halo time-course. Aliquots of intracellular material from each time point were resolved electrophoretically under nonreducing conditions and western blots were directly analyzed by fluorography for TMR ligand, to label selectively newly made proteins. See legend to Fig. EV1B, C for further details. The experiments were repeated three times with similar results. (C) Proposed model of pentameric IgM polymerization. Five $\mu_2L_2$ subunits bind covalently, forming the amyloid core. Then, an unstructured JC is incorporated which extends the amyloid core. Source data are available online for this figure.

pentamers, outrunning H-Cμ234tp2 or $\mu_2L_2$ subunits in completing a hexameric structure (Fig. 2C). Taken together, the above findings suggested that features of pentameric IgM promote stable interactions with JC.

## JC outcompetes the sixth IgM subunit during assembly also in vitro

Next, we reconstituted the process in the test tube using purified JC and the μ chain C-terminal domain, including the tailpiece (Cμ4tp), previously shown to be the minimal unit forming hexamers in vitro (Pasalic et al, 2017). In Cμ4tp, only C575 in the μtp is available for intermolecular bonds. Thus, no covalent polymers but only Cμ4tp$_2$ species are detectable in nonreducing SDS–PAGE (Pasalic et al, 2017). However, non-covalent assemblies of six Cμ4tp$_2$ dimers ((Cμ4tp$_2$)$_6$) formed in vitro can be readily detected by analytical size exclusion chromatography (SEC). (Cμ4tp$_2$)$_6$ eluted as a sharp peak (Fig. 3A, peak 1). Additionally, some Cμ4tp oxidation side products resulting in larger covalent oligomers were detected (Fig. EV2A, black insert boxes). Importantly, the addition of JC to the reaction caused a dramatic assembly shift: (Cμ4tp$_2$)$_6$ were no longer formed, but slower eluting species appeared whose size and composition are consistent with (Cμ4tp$_2$)$_5$ pentamers bound to a single JC (Fig. 3A peak 2). Titration experiments (Fig. EV2B) revealed that JC induced pentamer formation already at stoichiometric concentrations (0.1 mg/mL). JC caused the formation of side products (peak marked with *) if present in excess (>0.1 mg/mL). Assemblies with aberrant stoichiometries may form at the high concentrations reached in test tubes or in non-lymphoid transfectants. For further experiments, we chose ratios of Cμ4tp and JC, resulting in the highest amount of (Cμ4tp$_2$)$_5$-JC, while keeping amounts of hexamer and shoulder side products as low as possible (Fig. EV2B). JC alone eluted in a broad peak indicative of an ensemble of conformers (Fig. EV2C), consistent with its behavior in cells (Fig. 1D). SDS–PAGE analyses of individual fractions obtained after preparative HPLC-SE (Fig. EV2A) detected exclusively Cμ4tp$_2$ dimers in the hexamer fractions (1–2), while the pentamer fractions (3–5) contained a species with the mobility of a complex consisting of a JC linked to two Cμ4tp subunits (Fig. EV2A, lanes 3–5, red box). Reduction of all fractions (Fig. EV2D, top and bottom panel) resulted in Cμ4tp monomers. As expected, JC was found in fractions 3–5 (pentamers), fractions 5–8 (* side products), and fraction 9 (unbound JC co-eluting with Cμ4tp dimers). Taken together, these results established Cμ4tp and JC as the minimal

requirements for pentamer formation. They also recapitulated the results obtained in living cells, where JC shifted IgM polymerization toward pentamers at the expense of hexamers, confirming that JC is responsible for pentamer formation. Kinetic experiments (Fig. 3B–D) revealed that the presence of JC accelerated the formation of IgM complexes in vitro (Fig. 3B). In the absence of JC, Cμ4tp monomers transition to dimers and then to hexamers (Fig. 3C). Addition of JC (Fig. 3D) did not cause the accumulation of Cμ4tp-JC intermediates, confirming the results obtained in cells (Fig. 2A,B). These data highlight the striking similarity of polymerization in vivo and in vitro, confirming that JC selectively binds to oligomeric IgM. Interestingly, when added at the final stage of Cμ4tp assembly, JC did not significantly decrease the amount of preformed (Cμ4tp$_2$)$_6$ hexamers (Fig. EV2E), emphasizing the importance of timing in IgM assembly. The few pentamers detected at the end of the incubation were most probably formed by the remaining unassembled Cμ4tp dimers. Thus, the interaction of JC with nascent IgM complexes is timed to occur specifically before hexamer formation.

Next, we performed limited proteolysis experiments (Fig. 3E,F) to determine whether JC integration into IgM assemblies is accompanied by conformational changes that result in partial protection against proteolytic cleavage. To this end, Cμ4tp monomers and JC as well as HPLC-SE fractions containing hexamers or pentamers (fractions 2 and 4 from Fig. EV2A, respectively) were incubated with proteinase K (PK) for different times and the effect of proteolysis was analyzed by SDS-PAGE. Confirming previous results in living cells (Mancini et al, 2000), JC was rapidly and completely digested (Fig. 3E, see asterisks). Similarly, Cμ4tp monomers were cleaved, yielding a resistant fragment. In contrast, Cμ4tp hexamers (peak 1) displayed a higher PK resistance than the monomers (Fig. 3F). Cμ4tp$_2$, derived from non-covalent hexamers, were still detectable after 90 min of incubation with PK, and slowly converted into monomers thereafter. A similar effect was observed in the presence of JC (Fig. 3F). Cμ4tp$_2$ and Cμ4tp$_2$JC, the additional species present in the pentamer (Fig. EV2A, red box), were slowly processed by PK. These data indicated that JC is vulnerable to PK before assembly but becomes protected as part of the ß-sheet core that stabilizes IgM polymers (Fig. 3G).

To gain insight into the structure and disulfide bonding of recombinant JC, we performed circular dichroism (CD) and cross-linking mass spectrometry (MS). The CD spectrum of JC is consistent with that of a largely unstructured protein (Fig. 4A).

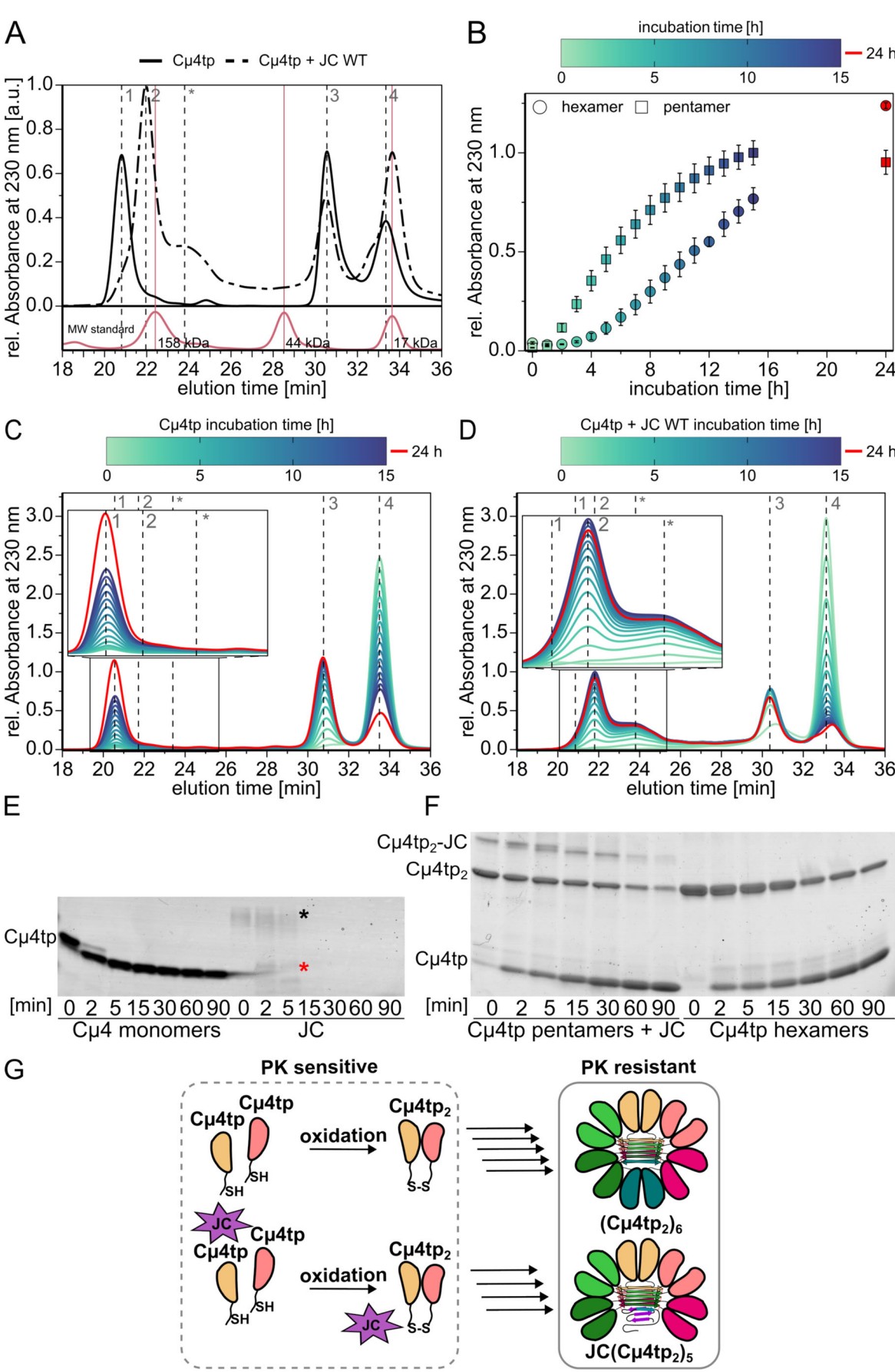

**Figure 3.  JC drives the formation of pentameric IgM in vitro.**

(A) Preferential binding of JC to $(C\mu4tp_2)_5$ pentamers in vitro. SEC chromatograms of recombinant $C\mu4tp$ (1 mg/mL) incubated in vitro with (dashed line) or without recombinant JC (0.5 mg/mL) (solid line) at room temperature for 24 h. In these conditions, $C\mu4tp$ monomers (peak 4, calculated mass: 14.5 kDa) oxidize to dimers (peak 3, calculated mass: 29 kDa), which further oligomerize to form hexamers (peak 1, calculated mass: 174.1 kDa, solid line). The addition of JC induces the formation of pentamers (peak 2, calculated mass: 160 kDa, dashed line), competing with the formation of hexamers. The chromatogram of the molecular mass standards (bovine γ-globulin, 158 kDa; chicken ovalbumin, 44 kDa; horse myoglobin,17 kDa) is shown in red. The complete chromatogram of the molecular mass standard is provided in Appendix Fig. S2A. All chromatograms were recorded thrice. (B) JC enhances the kinetics of oligomer formation in vitro. In the presence of JC (squares), oligomers are formed faster than in its absence (circles). After 15 h of incubation, pentamers are almost completely assembled, while hexamer formation by $C\mu4tp$ is still ongoing. Overall, therefore, the assembly of oligomers is faster in the presence of JC. The curves and standard deviations (black error bars) are derived from triplicates of the data shown in panels (C, D) below. (C, D) Kinetic measurement of hexamer formation. $C\mu4tp$ (1 mg/mL) incubated in PBS pH 7.4 at 25 °C forms hexamers over time (Pasalic et al, 2017). Monomers (peak 4) are gradually oxidized to dimers (peak 3), which proceed to assemble into hexamers (peak 1) (panel C). In the presence of JC (0.5 mg/mL), pentamers (peak 2) and side products (* peak) are formed (panel D), and no other intermediates of assembly are visible overtime. Measurements by HPLC-SE were performed every hour for 15 h and after 24 h h a final chromatogram was recorded. The experiments were performed in technical triplicates. (E, F) Oligomerization protects µtps and JC from proteolysis. Recombinant JC, $C\mu4tp$ monomers, $(C\mu4tp_2)_6$ hexamers or $(C\mu4tp_2)_5$-JC (fractions 2 or 4 Fig. EV2A, respectively) were incubated with Proteinase K and samples were taken at the indicated time points (minutes). The experiment was performed once. When not part of a complex, $C\mu4tp$ and JC are rapidly degraded (panel E). Already after 2 min, most $C\mu4tp$ is shortened to yield a resistant fragment, while JC (black asterisk) is completely degraded into small unstable fragments (red asterisk). In contrast, $C\mu4tp2$ and $C\mu4tp2$-JC (panel F) complexes that were part of hexamers or pentamers were largely protected from degradation. (G) Schematic representation of the main assemblies formed in our in vitro experiments. Before polymerization, JC and $C\mu4tp$ are sensitive to PK. The formation of hexamers or JC-containing pentamers confers partial protection. Owing to the absence of covalent bonds between two $C\mu4$ domains (apart from the inter-subunit ones between C575 in the tailpieces), hexamers and pentamers yield mainly $C\mu4tp_2$ dimers or $C\mu4tp_2$-JC complexes in denaturing SDS–PAGE. Source data are available online for this figure.

Analysis by cross-linking MS yielded a 100% peptide coverage for JC and revealed a mixture of non-native and native disulfide bonds (Fig. 4B). A few cysteines were shown to be involved in homomultimeric links, which proves the presence of oligomeric JCs. Thus, it cannot be excluded that a portion of the remaining identified bonds stems from multimeric JC. However, the amount of multimeric JC species should be small compared to the monomer as SE-HPLC analysis of JC (Fig. EV2C) clearly shows that the shoulder of higher molecular weight species is by far less intense than the broad monomeric JC peak. Similarly, nonreducing SDS-PAGE shows mostly monomeric species of JC (Appendix Fig. S2B). Interestingly, the numbers of peptide-to-spectrum matches (PSM counts) (Fig. 4C) are the highest for the crosslink between Cys92 with Cys102, which could be due to susceptibility of this site to protease cleavage. These results are consistent with their smeary appearance in nonreducing SDS gels (Fig. 1D, lane 5 compare top and bottom panel). Together with the results obtained in cells (Mancini et al, 2000), the above in vitro findings confirm that, on its own, JC does not adopt a unique structure with a defined disulfide bond pattern.

## Role of ERp44 in the formation of IgM polymers

In living cells, ERp44 patrols IgM hexamerization, recognizing non-native disulfides (Anelli et al, 2007; Giannone et al, 2022). To test whether ERp44 controls JC assembly into IgM pentamers as well, we reconstituted this process in ERp44$^{KO}$ HeLa cells (Fig. 5). Unexpectedly, JC was secreted by these cells. Under nonreducing conditions, secreted JC migrated as a continuous smear (Fig. 5A, lanes 4 and 7), which collapsed into a single band upon reduction with DTT (Fig. EV3A), suggesting that it consists of an ensemble of JC molecules containing a variety of intra- and inter-chain disulfide bonds. The expression of wild-type Halo tagged ERp44 (H-ERp44), but not of a mutant lacking the RDEL motif, which is abundantly secreted (Fig. 5C), efficiently restored JC retention (Fig. 5A, compare lanes 4-5-6). Upon expression of WT ERp44, JC accumulated in the lysates of ERp44$^{KO}$ HeLa, concentrating in numerous bands under nonreducing conditions (Fig. 5B, lanes 5 and 8). Most of these bands, but not all, contained ERp44 (Fig. 5D, see

asterisks). Thus, ERp44 prevents JC secretion and induces its binding to other proteins, that are presently not identified. ERp44 bound JC via C29 in its active CRFS site: upon C29S mutation, JC no longer formed complexes with ERp44 (Fig. EV3B). As expected, the secretion of unpolymerized IgM subunits by ERp44$^{KO}$ HeLa was prevented by ERp44 overexpression. In addition, fewer JC-containing pentamers were secreted (Fig. EV3C,D), consistent with the role of ERp44 in IgM-JC quality control (Giannone et al, 2022; Anelli et al, 2003).

Taken together, these findings indicate that JC can escape primary quality control in the ER of HeLa cells and form an ensemble of variants with different disulfide bonding patterns. ERp44 attacks non-native disulfide bonds surveying and rectifying IgM assembly.

## Conserved hydrophobic residues guide the insertion of JC into nascent pentamers

The finding that JC stably binds only pentamers raised questions as to how JC discriminates between these and smaller assemblies, and what triggers JC insertion. Hints came from the structural asymmetry and special arrangement of the five $\mu_2L_2$ subunits in the cryo-EM structure of IgM pentamers (Li et al, 2020a; Lyu et al, 2023). Curiously, the $\mu_2L_2$ subunit opposite to the JC (subunit 1 in Fig. 1A; Appendix Fig. S3A) displays its two µtps arranged in antiparallel orientations, each belonging to one distinct β-sheet of the core sandwich structure. In subunits 2–5, instead, the µtps are arranged in a parallel fashion and are part of either the top or the bottom β-sheet. In mature IgM pentamers (Li et al, 2020a), JC caps the ß-sandwich structure formed by the µtps in growing pentamers, adopting an arrangement that mimics subunit 1, with two β-strands (β3 and β4), pointing in opposite directions, that are interacting with the µtps of subunits 4 and 5 (Appendix Fig. S3A). We noted that the exposed hydrophobic core of the µtps in the IgM pentamer includes residues V564, L566, and M568 from subunits 4 and 5 (Fig. 6A; Appendix Fig. S3B). JC binding at this site leads to the extension of the β-sheet hydrogen bonding pattern, and the formation of hydrophobic contacts with several JC residues. Three highly conserved residues, namely I40 from JC strand β3 and F61 and Y63 from strand β4, which are located centrally in the

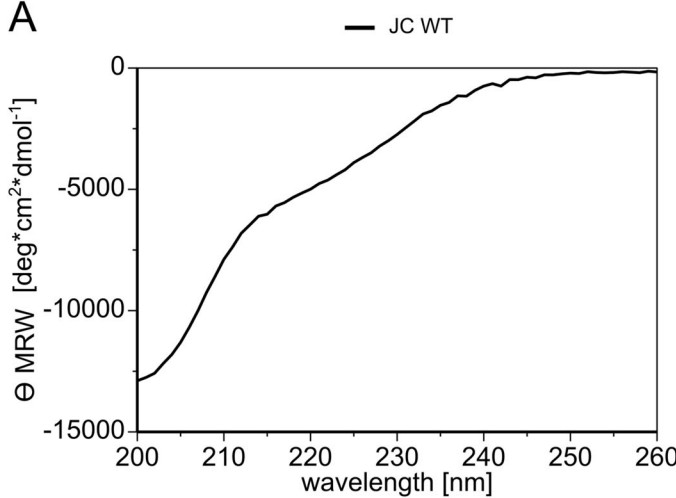

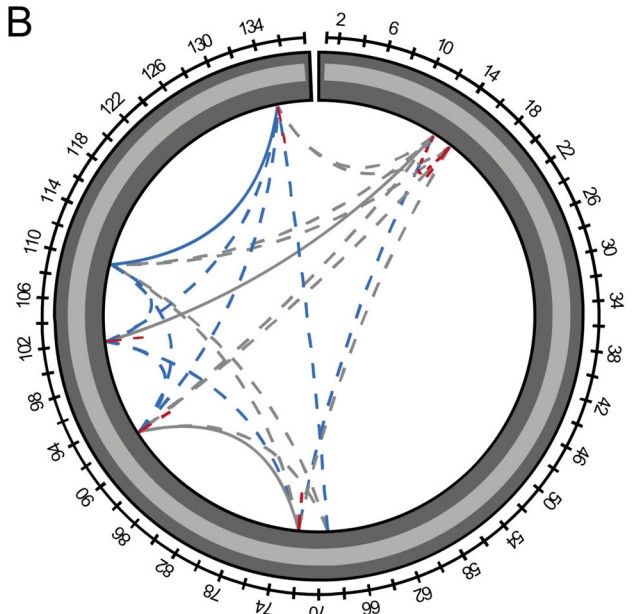

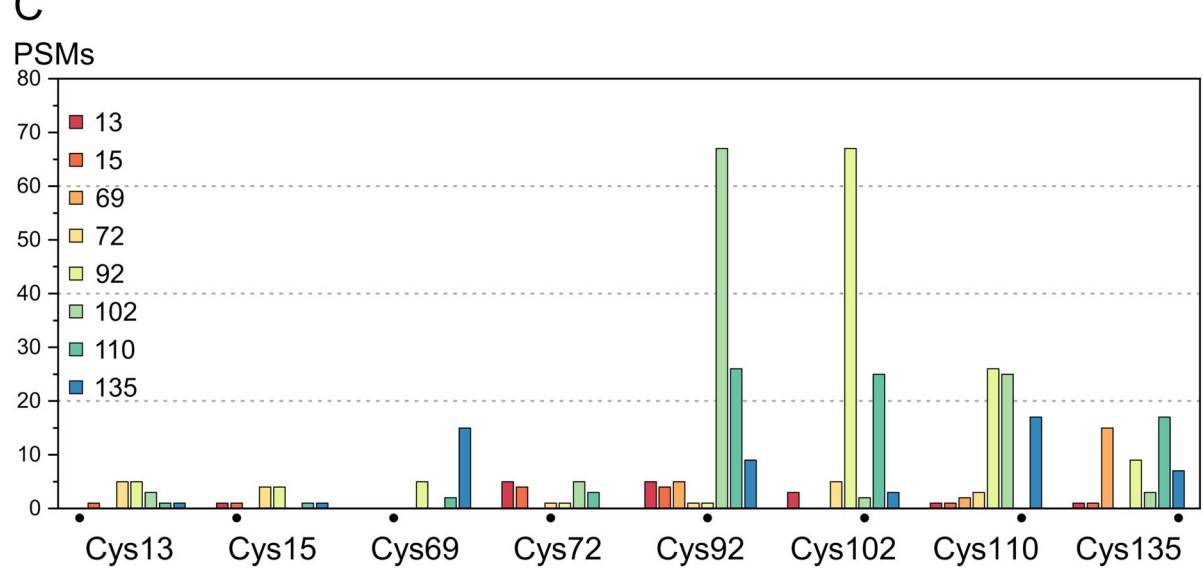

◀ **Figure 4. Recombinant JC is largely unstructured and forms native and non-native disulfides.**

(A) CD spectrum of refolded and oxidized recombinant JC. The pattern obtained by CD spectroscopy shows that JC is largely unstructured. The CD was performed once. (B, C) Cross-linking mass spectrometry analysis of disulfides formed in recombinant JC. Cross-linking MS analysis of JC disulfides revealed the presence of both native and non-native disulfides as well as intermolecular links in the recombinant protein. The protein coverage was 100%. The bar graphs shown in (C) summarize the PSM counts for recombinant JC. Note that the Cys92-Cys102 has higher PSM levels, likely reflecting the susceptibility of this site to protease cleavage. Source data are available online for this figure.

interaction surface, display shape complementarity to form a tightly packed hydrophobic core (Fig. 6A). Therefore, I40, F61 and Y63 might be important in the encounter complex between JC and the core of nascent pentameric IgM. To test this, we co-expressed JC mutants in which these residues were mutated to alanine (I40A, F61A and Y63A, Fig. 6B–F) or serine (I40S, F61S and Y63S, Fig. EV4A,B) with H-Cμ234tp in HEK293T cells (Figs. 6B,C and EV4A,B) or in HeLa-ERp44^KO cells (Figs. 6D,E and EV4C,D). JC binding to IgM complexes was monitored by the ability to prevent hexamer formation. IgM assembly in the presence of the three JC mutants generated pentamers which were secreted poorly, although for different reasons. Mutating I40 prevented JC insertion (Fig. 6B,C, compare lanes 7-8). The phenotype was more pronounced for I40S than for I40A, in line with the notion that hydrophobic packing stabilizes the core structure (Fig. EV4A,B). Conversely, the F61A and Y63A mutants formed abundant polymers (Fig. 6C, lanes 4-5), which, however, were largely retained by HEK293 cells (Fig. 6C, lanes 9-10). The failure of JC hydrophobic mutants to correctly bind pentameric IgM leads to the secretion of hexamers only (Figs. 6B and EV4A, lanes 9-10). However, pentamers containing JC hydrophobic mutants were secreted by HeLa-ERp44^KO cells (Fig. 6D,E, lanes 3, 4, and 5). Retention was rescued by ERp44 expression (Figs. 6D,E, lanes 7-8-9 and EV4C,D), confirming its pivotal role in IgM biogenesis. These results point to stringent cellular quality control by ERp44 on these misassembled polymeric IgM.

In vitro, SEC analyses of interaction yielded results comparable to those obtained in cells (Fig. 6F). The I40A JC mutant was unable to shift the (Cμ4tp₂)₆ hexamer towards pentamer formation. A small hexamer to pentamer shift was induced by the F61A and Y63A mutants: however, both mutants formed wider peaks with earlier elution times than those containing WT JC, suggesting altered structures (Fig. 6F). These results strengthen the observation made in living cells, in which F61 and Y63 containing polymers were formed but retained by ERp44. Taken together, these findings suggest that I40, F61 and Y63 are essential for the initial and correct interactions of JC with the core of the nascent pentameric structure. F61 and Y63 may promote the conformational changes needed to build secretable JC-containing pentamers.

## Disulfide rearrangements drive JC folding and covalent pentamer binding

The above results highlighted the key roles of the hydrophobic encounter complex in JC insertion, and of ERp44 in patrolling assembly. Based on the above results, IgM binding is likely initiated by interactions of I40, F61, and Y63 with the μtps of subunits 4 and 5 exposed by nascent IgM pentamers in opposite directions. Upon docking, JC needs to covalently bind the two μtps via C15 and C69 and adopt its final structure with three intra-chain disulfides (C72–92, C13–102, and C110–135). To investigate the underlying

mechanisms, we generated cysteine replacement mutants and tested their capability to form pentamers in cellula (Figs. 7A,B and EV5A) and in vitro (Figs. 7C,D and EV5B).

In cells, both C15S and C69S variants yielded secretion-competent pentamers, albeit to a lower extent for C15S than WT JC (Fig. 7A lanes 3 and 6). Surprisingly, also their simultaneous replacement did not completely abolish the covalent binding of the double mutant C15-69S to IgM (Fig. 7A, lane 7), suggesting that other JC residues might interact with C575 in the μtp. While the C13S single mutant was partially incorporated into secreted IgM, the C13-C15S double mutant was not (Figs. 7A, lane 8-9 and EV5A lanes 8-9). Accordingly, cells expressing C13-15S secreted abundant hexamers (Fig. 7B), confirming their poor insertion into IgM. In contrast, the mutants C72S and C69-C72S were inserted in polymeric IgM and partly secreted, with a lesser degree for the latter one (Fig. 7A lanes 4-5). Taken together, these results point to a pivotal role of the C13-C15 couple in triggering IgM binding.

In vitro, the mutants C13S and C15S were able to integrate to some extent into IgM pentamers, albeit displaying varying distribution of different species (Fig. 7C). Shoulder formation or shifts towards the hexamer peak in SEC analyses indicate the formation of a mixture of hexamers, pentamers in different ratios, while tailing hints at the presence of other undefined assemblies. The tendency of the C15S mutant towards the hexamer shoulder implies a pivotal role of this residue. More intermediates eluting after the pentamer were formed upon mutation of either C69 or C72 (Fig. 7D). The double mutants C69-72S (Fig. 7D) or C13-15S (Fig. 7C) formed larger amounts of hexamers than the single point mutants did. In the former, the pentamer peak is visible as merely a slight shoulder.

These results imply that in vitro both the C13xC15 and C69xxC72 motifs are important for pentamer assembly. However, the removal of both cysteines, which bind to IgM (C15 and C69), had only mild effects (Fig. 7D). A pronounced hexamer shoulder was evident, but despite a slight shift, still plenty of pentamers were formed in the presence of the C15S-C69S double mutant, confirming our in cellula results (Fig. 7A,B). In conclusion, in vitro, at least one cysteine of both the CxC and CxxC motifs needs to be present for JC binding to nascent IgM complexes.

We also analyzed two JC deletion mutants lacking hairpins that establish van-der-Waals contacts with the Cμ4 domains (Li et al, 2020b; Kumar et al, 2020). The C108-C135 disulfide stabilizes a loop crucial for IgM interaction with the pIgR and the secretory component but not for binding IgA (Johansen et al, 2001). Accordingly, a mutant in which this region was deleted (ΔHP3, Δ108-139) retained most of its capability to form pentamers in vitro (Fig. EV5B). The presence of a hexamer shoulder, however, shows that the region contributes to efficient assembly with pentamers. In contrast, the mutant ΔHP2 (Δ72-92) formed large amounts of aberrant assemblies. We conclude that JC incorporation

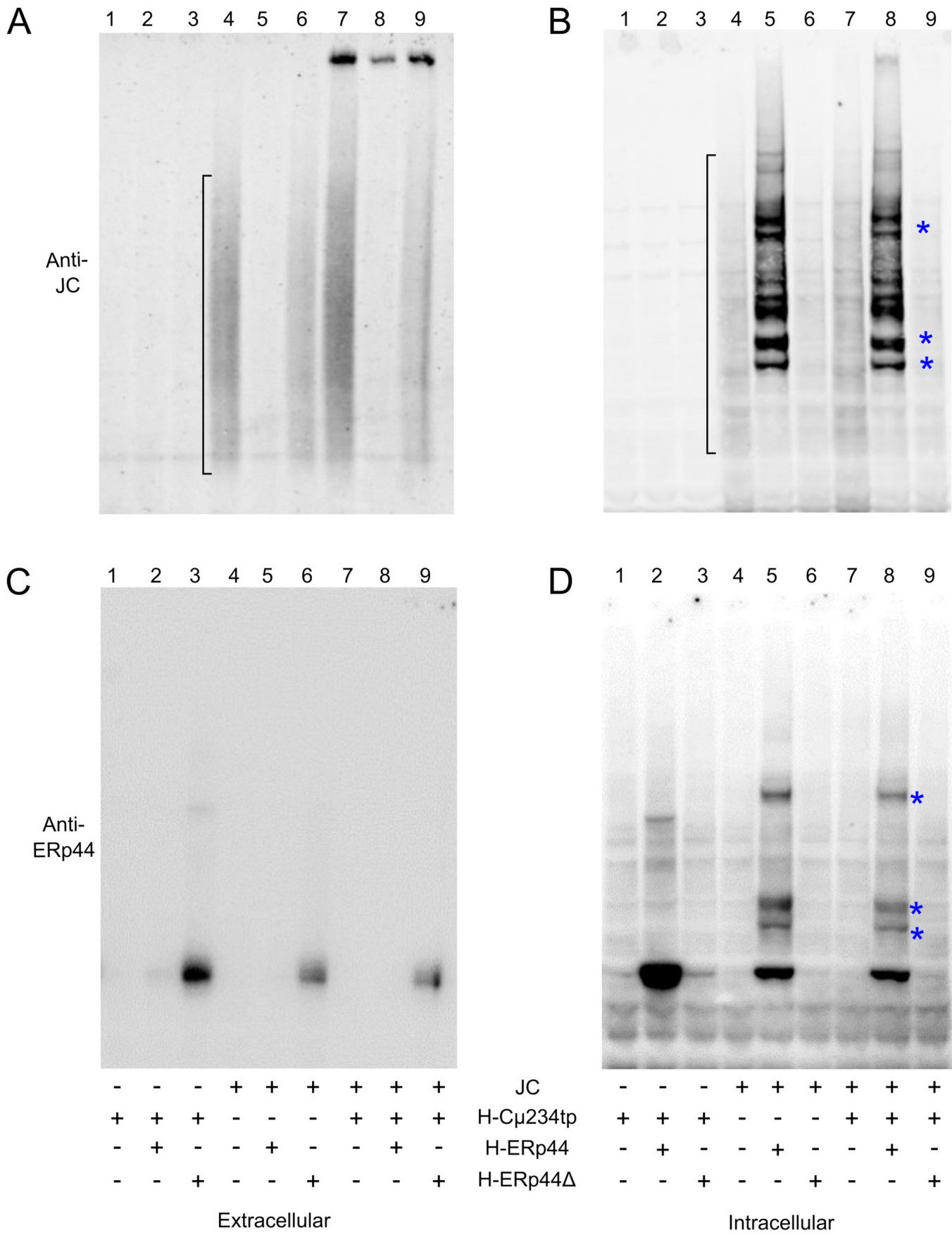

**Figure 5. Pivotal role of ERp44 in preventing the secretion of unassembled JC.**

HeLa-ERp44$^{KO}$ cells were transfected as indicated, and aliquots from their supernatants (**A, C** Extracellular) or lysates (**B, D** Intracellular) resolved under nonreducing conditions and decorated with anti-JC or anti-ERp44, as indicated. The black brackets point at the continuous smear of JC that are secreted by (**A**) cells lacking ERp44. Clearly, the presence of ERp44 prevents the secretion of free JC. Blue asterisks point at JC-ERp44 covalent complexes (**B**). ERp44ΔRDEL (lanes 3, 6, 9) is secreted (**C**) and fails to form covalent complexes with JC (**D**). The experiment was repeated at least three times with similar results. Source data are available online for this figure.

strongly relies on cysteines 13, 15, 69, and 72, but the contacts between JC hairpins and the Cμ4 domains are of additional significance, as their absence induces the increased formation of either side products (ΔHP2) or hexamers (ΔHP3) in vitro (Johansen et al, 2001).

In summary, our data suggest that an ordered sequence of events dictates the biogenesis of JC-containing IgM pentamers in cellula as well as in vitro. Once the hydrophobic cues (revealed by the I40, F61 and Y63 mutants) have placed JC in the proper orientation to extend the amyloid core, disulfide rearrangements are triggered that lead C15 and C69 to form stable bonds with the two μtp C575, properly positioned in nascent IgM pentamers. Since the simultaneous absence of C13 and C15 was almost fatal for pentamer formation both in cellula and in vitro, and intra-chain bonds between these residues were found by MS in vitro (Fig. 4B), we surmise that the two cysteine residues in the CxC motif be necessary for starting the covalent binding of JC into IgM nascent pentamers. Once the process is initiated, the native intra-chain bonds as well as interactions with the hairpins 2 and 3, likely stabilize the final JC conformation.

## Discussion

Our findings explain how IgM are produced with five μ$_2$L$_2$ subunits and one JC, a stoichiometry that confers competence for transcytosis without major loss of avidity. The solution lies in the delayed folding of JC. Their small size and flexibility favor insertion at the expense of a sixth μ$_2$L$_2$ subunit in the formation of secretion-competent hexameric assemblies. Indeed, the angle left for the sixth unit in nascent pentamers is ≤60° (Hiramoto et al, 2018), likely conferring an advantage to the small JC.

JC inserts into growing IgM oligomers following similar rules both in vitro and in cellula. Thus, the process is governed by the intrinsic properties of these proteins. JC and μtp are extremely conserved from sharks to humans (Lyu et al, 2023; Marchalonis et al, 1993; Castro and Flajnik, 2014), yielding an assembly module that endured hundreds of millions of years. Protease sensitivity, circular dichroism, the richness in diverse non-native disulfides, and their rapid degradation concur in putting JC in the family of intrinsically disordered proteins. Linking JC folding to structural instructions present exclusively in nascent pentameric IgM allows the specific formation of complexes comprising five μ$_2$L$_2$ subunits and a single JC. Such assemblies satisfy the important evolutionary pressure of creating transcytosis-competent IgM whilst maintaining high avidity for antigen and complement. Since a single JC is sufficient to interact with pIgRs, we surmise that a 5:1 configuration would be the evolutionary optimum for these two biological functions, compared to 4:2 or 3:3 arrangements, as in the 5:1 arrangement the maximum number of antigen binding units is preserved. Delayed folding is probably important to allow JC to

outcompete the sixth μ$_2$L$_2$ subunit in the tight space offered by a growing pentamer. Thus, the rigid and bulkier μ$_2$L$_2$ subunits are likely slower than flexible JC in filling the available space.

Based on our results, we propose a mechanism in which the five μ$_2$L$_2$ subunits that form a nascent pentamer present two μtps in an arrangement suitable for JC docking. At this step, the small and structurally flexible JC accesses a nascent pentamer forming an encounter complex based on hydrophobic interactions between conserved aminoacids in the μtps and the β3 and β4 strands of JC. It remains to be seen whether partial structuring of the JC β3 and β4 strands precedes, and perhaps drives, the interactions with IgM. The establishment of these initial interactions triggers disulfide rearrangements that eventually lead to the formation of the final bonds. Our cysteine mutagenesis screens suggest a pivotal role for the C13-C15 couple in triggering disulfide reshuffling. In support of this is also the existence of a non-native disulfide between C13 and C15 identified by mass spectrometry analyses. Key in the preferential recruitment of a JC is probably also the arrangement of the μtps β-strands exposed by nascent pentamers.

An unexpected finding in this study was the secretion of JC as a mixture of isoforms with non-native intra- and inter-chain disulfides by cells lacking ERp44. Thus, HeLa cells rely mainly on ERp44 to weed out assemblies with wrong conformation(s). JC eludes primary ER quality control (Reddy et al, 1996) and reaches the subregion patrolled by ERp44. Additional factors may assist and fine-tune IgM assembly in plasma cells, including Perp1, also known as MZB1 or CNPY5 (Sowa et al, 2021; van Anken et al, 2009; Shimizu et al, 2009; Anelli and van Anken, 2013) whose expression is thought to improve JC binding to IgA (Wei and Wang, 2021).

It is of note how non-native disulfides emerge as fundamental intermediates in the process of IgM polymerization. A non-native intra-subunit bond between two C575 residues within a single μ$_2$L$_2$ subunit is important to trigger IgM hexamerization (Pasalic et al, 2017; Giannone et al, 2022). A reservoir of intra-chain disulfide bonds in the unfolded-unassembled JC may facilitate the interchange reactions that take place in the core of nascent polymers. ERp44 may promote polymerization favoring the retrieval of aberrant complexes and/or the co-localization of JC and μ$_2$L$_2$ in a suitable sub-compartment.

In conclusion, our work defines the basic steps of how the immune system can produce two types of high-avidity IgM oligomers with different biological functions. The association-triggered fit of JC calls to mind the folding of the IgG C$_H$1 domain upon association of heavy and light chains (Feige et al, 2009) offering an amazing mechanistic similarity between humoral and mucosal immunity. To promptly combat diverse pathogens with limited collateral damage, cells of the B lineage evolved sophisticated mechanisms driving the correct association and quality control of their complex molecular arsenal.

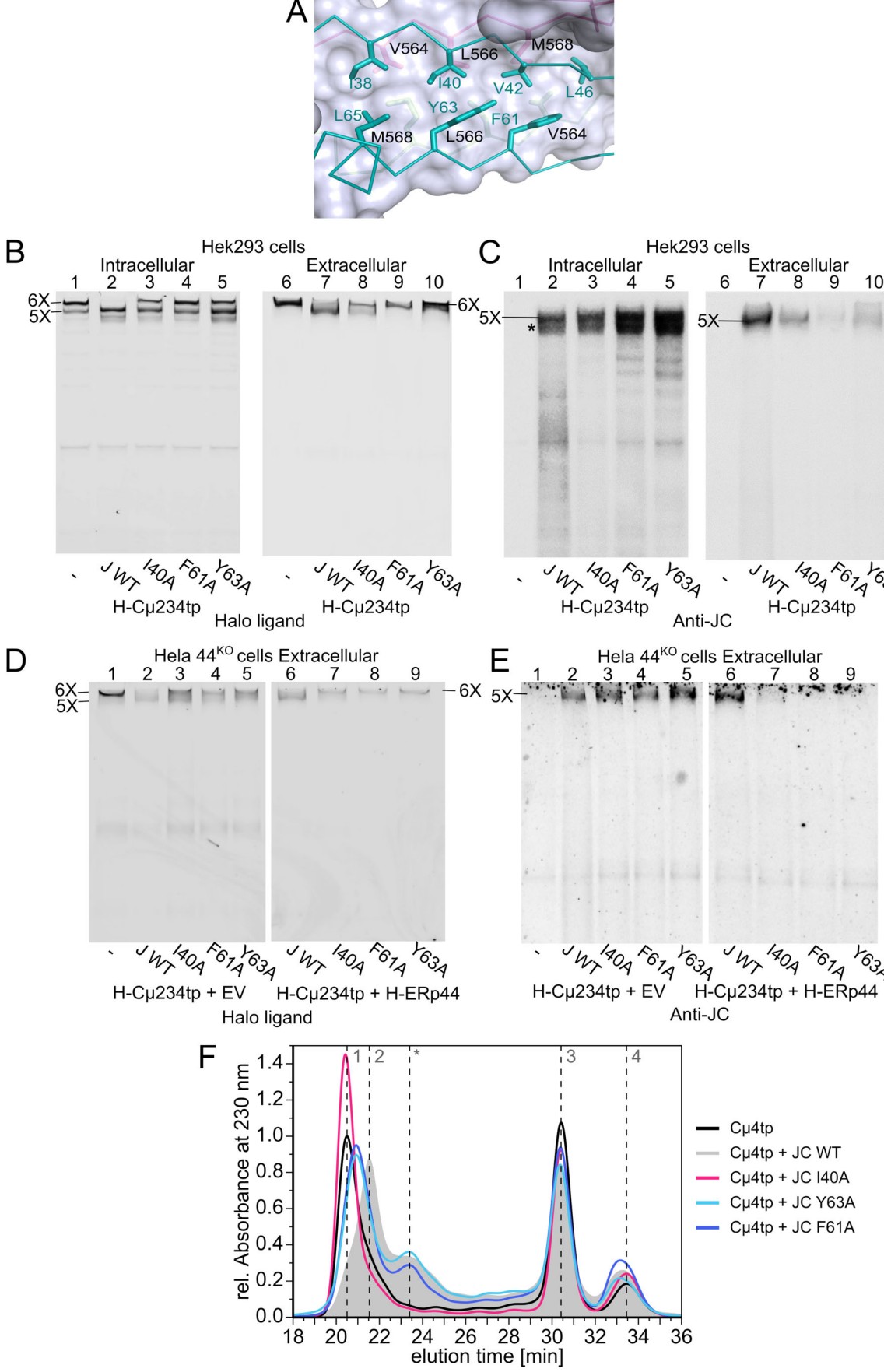

**Figure 6. Hydrophobic residues in JC guide the interaction with nascent pentameric IgM.**

(A) Contacts between JC and the µtps of subunits 4 and 5. The vicinity of JC I40, F61, and Y63 with V564, L566, and M568 in the µtps of subunits 4 and 5 points at a potential role of these residues in polymerization. The images were generated using PyMOL (The PyMOL Molecular Graphics System) and are extrapolated from the Cryo-EM structure of human IgM-Fc in complex with the J chain and the ectodomain of pIgR. (B, C). Different roles of conserved hydrophobic aminoacids in JC binding to nascent pentamers. The lysates and supernatants of HEK293 transfectants co-expressing Halo-Cµ234tp and WT, mutated (I40A, F61A, and Y63A) or no JC, were decorated with Halo ligands (B) or anti-JC (C), as indicated. Note that only WT JC efficiently prevents the formation of hexamers (6x), favoring lower-order assemblies. Unexpectedly, F61A and Y63A mutants associated more with intracellular H-Cµ234tp polymers than WT (lanes 4-5, Panel C). However, they were barely secreted (lanes 9-10, Panel B, C). The experiments were performed at least three times with similar results. (D, E) ERp44 prevents the secretion of aberrantly assembled pentamers. Aliquots of the supernatants from the indicated HeLa-ERp44$^{KO}$ cells transfectants were decorated with Halo ligands (D) or anti-JC (E). Note that upon ERp44 expression, only hexamers are secreted by cells expressing mutant JC, consistent with their failure to form native pentamers. The experiments were repeated at least three times with similar results. (F) Aberrant in vitro interactions of JC lacking hydrophobic residues with Cµ4tp. The HPLC-SE profiles of JC-Cµ4tp were obtained using a Superdex 200 10/300 GL column in PBS at 25 °C. The JC mutants I40A (pink profile), F61A (dark blue trace), and Y63A (light blue profile) were incubated with Cµ4tp for 24 h at room temperature at a concentration of 0.5 and 1 mg/mL, respectively. A sample volume of 10 µL was injected. The profile of Cµ4tp-JC WT, represented by a gray background, was used as a reference. The experiments were performed in technical triplicates. Source data are available online for this figure.

# Methods

### Reagents and tools table

| Reagent/resource | Reference or source | Identifier or catalog number |
|---|---|---|
| **Experimental models** | | |
| *E.coli* BL21 (DE3) Competent cells | New England Biolabs | Cat #C25271 |
| *E. coli* XL1-Blue | Agilent | Cat # 200249 |
| NS0 | Galfrè and Milstein, 1981 | N/A |
| J558 | Oi et al, 1983 | N/A |
| J[µ$_s$] | Sitia et al, 1990 | N/A |
| Nµ1 | Sitia et al, 1990 | N/A |
| HeLa ERp44KO | Giannone et al, 2022 | N/A |
| HEK293 | Giannone et al, 2022 | N/A |
| **Recombinant DNA** | | |
| Plasmid: pET28b_mJC_WT | Thermo Fisher Scientific | IGJ_MOUSE; P01592; residues 22-159; M on position 1 |
| Plasmid: pET28b_Cµ4tp_WT | Thermo Fisher Scientific | P01872; IGHM_MOUSE; E446–Y576 |
| Plasmid: pET28b_mJC_C13S | This paper | N/A |
| Plasmid: pET28b_mJC_C13S_C69S | This paper | N/A |
| Plasmid: pET28b_mJC_C15S | This paper | N/A |
| Plasmid: pET28b_mJC_C69S | This paper | N/A |
| Plasmid: pET28b_mJC_C72S | This paper | N/A |
| Plasmid: pET28b_mJC_C13S_C15S | Thermo Fisher Scientific | N/A |
| Plasmid: pET28b_mJC_C69S_C72S | Thermo Fisher Scientific | N/A |
| Plasmid: pET28b_mJC_ΔHP2 | This paper | N/A |
| Plasmid: pET28b_mJC_ΔHP3 | This paper | N/A |
| Plasmid: pET28b_mJC_I40A | This paper | N/A |
| Plasmid: pET28b_mJC_F61A | This paper | N/A |
| Plasmid: pET28b_mJC_F63A | This paper | N/A |

| Reagent/resource | Reference or source | Identifier or catalog number |
|---|---|---|
| Plasmid: pcDNA3.1_H-Cµ234tp | Giannone et al, 2022 | N/A |
| Plasmid: pcDNA3.1_JCmyc | This paper | N/A |
| Plasmid: pcDNA3.1 H-ERp44 | Mossuto et al, 2014 | N/A |
| Plasmid: pcDNA3.1 H-ERp44Δ | Anelli et al, 2003 | N/A |
| Plasmid: pcDNA3.1 H-ERp44C29S | Anelli et al, 2003 | N/A |
| **Antibodies** | | |
| Mouse monoclonal anti-ERp44 36C9 | Anelli et al, 2003 | N/A |
| Rabbit anti-J chain | Fagioli and Sitia, 2001 | N/A |
| Goat anti-mouse IgM (µ chain) Alexa fluor 647 | Life Technologies — Invitrogen | #A21048 |
| Goat anti-mouse lambda-HRP | Southern Biotech | #1060-05 |
| **Oligonucleotides and other sequence-based reagents** | | |
| Primer for pET28b_mJC_C13S_Forward Used for mutants pET28b_mJC_C13S and pET28b_mJC_C13S_C69S | Eurofins Genomics | CGATAACAAAtct ATGTGTACCCG |
| Primer for pET28b_mJC_C13S_Reverse Used for mutants pET28b_mJC_C13S and pET28b_mJC_C13S_C69S | Eurofins Genomics | GCCAGAATG GTTGCTTCA |
| Primer for pET28b_mJC_C15S_Forward | Eurofins Genomics | CAAATGTATGtct ACCCGTGTTAC |
| Primer for pET28b_mJC_C15S_Reverse | Eurofins Genomics | TTATCGGCCA GAATGGTTG |
| Primer for pET28b_mJC_C69S_Forward Used for mutants pET28b_mJC_C69S and pET28b_mJC_C13S_C69S | Eurofins Genomics | GTCCGATGTGt ccAAAAAATGTG |

| Reagent/resource | Reference or source | Identifier or catalog number |
|---|---|---|
| Primer for pET28b_mJC_C69S_Reverse Used for mutants pET28b_mJC_C69S and pET28b_mJC_C13S_C69S | Eurofins Genomics | AGATGATAAAC AAAAATTACGAC |
| Primer for pET28b_mJC_C72S_Forward | Eurofins Genomics | GTGCAAAAAA tctGATCCGGTTG |
| Primer for pET28b_mJC_C72S_Reverse | Eurofins Genomics | ACATCGGACA GATGATAAAC |
| Primer for pET28b_mJC_ΔHP2_(Δ73-93)_Forward | Eurofins Genomics | AATGAAGATG ATGGTGTTC |
| Primer for pET28b_mJC_ΔHP2_(Δ73-93)_Reverse | Eurofins Genomics | TTTTTTGCACA CATCGGAC |
| Primer for pET28b_mJC_ΔHP3_(Δ108-139): Forward: | Eurofins Genomics | TAATAAAAGC TTGCGGCC |
| Primer for pET28b_mJC_ΔHP3_(Δ108-139): Reverse: | Eurofins Genomics | ATCATACATAT AGCACGTTTC |
| Primer for pET28b_mJC_I40A: Forward: | Eurofins Genomics | CAATATTCGT gccGTTGTG CCGC |
| Primer for pET28b_mJC_I40A: Reverse: | Eurofins Genomics | CGTTCCACAA TATCTTCATTC |
| Primer for pET28b_mJC_F61A: Forward: | Eurofins Genomics | GCGTCGTAAT gccGTTTAT CATCTGTCCGA TGTG |
| Primer for pET28b_mJC_F61A: Reverse: | Eurofins Genomics | AGCGGAGAG GTCGGATCG |
| Primer for pET28b_mJC_Y63A: Forward: | Eurofins Genomics | TAATTTTGTT gcgCATCTGT CCGATGTGTGC |
| Primer for pET28b_mJC_Y63A: Reverse: | Eurofins Genomics | CGACGCAG CGGAGAG |
| Primer for pcDNA3.1 JC_C13S Forward: | Metabion International | CTTGCTGAC AACAAAGCCA RGTGTACCCG AGTTACC |
| Primer for pcDNA3.1 JC_C13S Reverse: | Metabion International | GGTAACTCGGG TACACATGGC TTTGTTGTCA GCAAG |
| Primer for pcDNA3.1_JC_F61A Forward: | Metabion international | CTGAGAAGGA ACGCTGTA TACCAT |
| Primer for pcDNA3.1_JC_F61A Reverse: | Metabion international | CAAATGGTATA CGGCGTTCCTT CTCAG |
| Primer for pcDNA3.1_JC_I40A Forward: | Metabion international | GAGAGAAATAT CCGAGCCGT TGTCCCTTTG |

| Reagent/resource | Reference or source | Identifier or catalog number |
|---|---|---|
| Primer for pcDNA3.1_JC_I40A Reverse: | Metabion international | CAAAGGGACAA CGGCTCGGATA TTTCTCTC |
| Primer for pcDNA3.1_JC_Y63A Forward: | Metabion international | AGGAACTTTGT AGCCCATTT GTCAGAC |
| Primer for pcDNA3.1_JC_Y63A Reverse: | Metabion international | GTCTGACAAAT GGGCTACAA AGTTCCT |
| Primer for pcDNA3.1_JC_F61S Forward: | Metabion international | CTGAGAAGG AACTCGGTA TACCAT |
| Primer for pcDNA3.1_JC_F61S Reverse: | Metabion international | ATGGTATACC GAGTTCCTT CTCAG |
| Primer for pcDNA3.1_JC_I40S Forward: | Metabion international | AGAAATATCCG ATCGGTTGTC CCTTTG |
| Primer for pcDNA3.1_JC_I40S Reverse: | Metabion international | CAAAGGGACAA CCGATCGGAT ATTTCT |
| Primer for pcDNA3.1_JC_Y63S Forward: | Metabion international | AGGAACTTTGT ATCGCATTTG TCAGAC |
| Primer for pcDNA3.1_JC_Y63S Reverse: | Metabion international | GTCTGACAAAT GGGATACAAAG TTCCT |
| **Chemicals, Enzymes and other reagents** | | |
| Halo-tag florescent ligand (R110Direct) | Promega | #G322A |
| Halo-tag florescent ligand (TMRDirect) | Promega | #G299A |
| jetPEI DNA Transfection Reagent | Polyplus | #101000053 |
| Nonidet P-40 | Fluka BioChemika | #74385 |
| N-ethylmaleimide (NEM) | Sigma-Aldrich | #E3876-5G |
| Protease inhibitor cOmplete Tablets | Roche | #04693116001 |
| Proteinase K (from Tritirachium album) | Merck | Cat #1245680500; CAS 39450-01-6 |
| Trypsin, Sequencing Grade | Merck | Cat #11418475001 |
| Asp-N, Sequencing Grade | Promega | Cat #V1621 |
| Q5® High-Fidelity DNA Polymerase | New England Biolabs | Cat #M0491S |
| DPNI | New England Biolabs | Cat #R0176S |
| T4 Polynucleotide Kinase | New England Biolabs | Cat #M0201S |
| T4 DNA Polymerase | New England Biolabs | Cat #M0203S |
| Deoxynucleotide (dNTP) Solution Mix | New England Biolabs | Cat #N0447S |
| Q5® Reaction Buffer | New England Biolabs | Cat #B9027S |
| T4 DNA Ligase Reaction Buffer | New England Biolabs | Cat #B0202S |

| Reagent/resource | Reference or source | Identifier or catalog number |
|---|---|---|
| Tris-(2-carboxyethyl)-phosphine hydrochloride (TCEP) | Carl Roth | Cat #HN95.4; CAS 51805-45-9 |
| 1,4-Dithiothreit (DTT) | Carl Roth | Cat # 6908.2; 3483-12-3 |
| Water ULC/MS - CC/SFC | Biosolve | Cat #232141; CAS 7732-18-5 |
| Acetonitrile | Merck | Cat #1.00030.2500; CAS 75-05-8 |
| 2-Iodoacetamide | Merck | Cat #804744; CAS 144-48-9 |
| Formic Acid For LC-MS LiChropur | Merck | Cat #5330020050; CAS 64-18-6 |
| PMSF | Merck | Cat #P-7626; 329-98-6 |
| Protease Inhibitor Mix HP | SERVA | Cat #39106.03 |
| Trizma® Base | Merck | Cat #T1503; CAS 77-86-1 |
| Titriplex® III (Ethylendinitrilotetraessigs äure Dinatriumsalz-Dihydrat) | Merck | Cat #1.08421.1000; CAS 6381-92-6 |
| Urea | Merck | Cat #1.08488.1000; CAS 57-13-6 |
| Sodium chloride | Carl Roth | Cat # 9265.1; CAS 7647-14-5 |
| di-Sodium hydrogen phosphate dihydrate | Merck | Cat #106580; CAS 10028-24-7 |
| Potassium dihydrogen phosphate | Merck | Cat # 104877; CAS 7778-77-0 |
| Potassium chloride | Roche | Cat #10174645001; CAS 7447-40-7 |
| L-Glutathione oxidized | Roche | Cat #105635; CAS 27025-41-8 |
| L-Glutathione reduced | Sigma | Cat #G4251; CAS 70-18-8 |
| LB Broth (Luria/Miller) | Carl Roth | Cat #X968.1 |
| Kanamycin | Carl Roth | Cat #T832.5; CAS 25389-94-0 |
| Hydrochlorid acid 32% | Merck | Cat #1.00319.2511; CAS 7647-01-0 |
| ortho-Phosphoric acid | Carl Roth | Cat #6366.2; CAS 7664-38-2 |
| Sodium hydroxide | Carl Roth | Cat #1CCX.1; CAS 1310-73-2 |
| Serva Blue G | SERVA | Cat #35050.03; 6104-58-1 |
| Bromophenol blue | Merck | Cat # 1.11746.0005; CAS 115-39-9 |
| 2-Mercaptoethanol | Carl Roth | Cat #4227.3; CAS 60-24-2 |
| SDS grained pure | Applichem | Cat #A7249,0500; CAS 151-21-3 |
| Glycin | Carl Roth | Cat #3908.3; CAS 56-40-6 |
| Glycerol | Carl Roth | Cat #6962.2; CAS 56-81-5 |
| Ethanol | Carl Roth | Cat #1HP8.2; CAS 64-17-5 |
| Acetic acid | Carl Roth | Cat #3738.5; CAS 64-17-5 |
| **Software** | | |
| Percolator (version 2.08) | Käll et al | https://github.com/percolator/percolator/releases |
| Thermo Scientific Foundation software version 3.1sp9 | Thermo Fisher Scientific | https://knowledge1.thermofisher.com/Software_and_Downloads/Chromatography_and_Mass_Spectrometry_Software/Xcalibur/Xcalibur_Operator_Manuals/XCALI-97619_-_Rev_A_-_Thermo_Foundation_Administrator_Guide_-Software_Version_3.1 |
| Xcalibur version 4.6 | Thermo Fisher Scientific | https://www.thermofisher.com/de/de/home/industrial/mass-spectrometry/liquid-chromatography-mass-spectrometry-lc-ms/lc-ms-software/lc-ms-data-acquisition-software/xcalibur-data-acquisition-interpretation-software.html |
| ProXL | Riffle et al | https://proxl-ms.org/ |
| ProteoWizard converter | Chambers et al | https://proteowizard.sourceforge.io/ |
| Kojak (version 2.0.0 alpha 22) | Hoopmann et al, Hoopmann et al | https://kojak-ms.systemsbiology.net/index.html |
| The PRoteomics IDEntifications (PRIDE) database | | https://www.ebi.ac.uk/pride/ |
| The ProteomeXchange consortium | | https://www.proteomexchange.org/ |
| Lab Solution/LC Solution 1.2 | Shimadzu | https://www.shimadzu.com/ |
| RStudio (2023.12.1 + 402) | R Core Team (2023) | https://www.R-project.org |
| R 3.3.0+ | R Core Team (2023) | https://cran.rstudio.com/ |
| Inkscape (1.3.2) | | gitlab.com/inkscape/inkscape |
| NEBaseChanger | New England Biolabs | https://nebasechanger.neb.com/ |

| Reagent/resource | Reference or source | Identifier or catalog number |
|---|---|---|
| Tm Calculator | New England Biolabs | https://tmcalculator.neb.com/#!/main |
| PyMOL 2.5 | Schrödinger, LLC | https://pymol.org/2/#page-top |
| **Other** | | |
| SERVAGel TG PRIME 4-20% | SERVA | Cat #43289.01 |
| Cuvette for CD UV/VIS (200 - 2500 nm), path length 1 mm | Hellma | Cat #110-1-40 |
| Superdex 200 Increase 10/300 SEC column | Cytiva | Cat # 28990944 |
| Q-Sepharose Fast Flow resin | Cytiva | Cat # 17051004 |
| HiLoad Superdex 75 pg 16 mm preparative size exclusion chromatography column | Cytiva | Cat # 28989333 |
| HiLoad Superdex 75 pg 26 mm preparative size exclusion chromatography column | Cytiva | Cat # 28989334 |

## Methods and protocols

### Cell lines

ERp44$^{KO}$ (Giannone et al, 2022) or WT HeLa cells, HEK293 cells from ATCC, NS0 (Galfrè and Milstein, 1981), and J558 (Oi et al, 1983) were cultured in Dulbecco's modified Eagle's medium (DMEM) supplemented with 10% fetal calf serum (FCS; Euro-Clone), penicillin-streptomycin (5 mg/ml) (Lonza), and glutamine (2 mM). About 10 µg/mL mycophenolic acid (MPA), 20 µg/mL hypoxanthine, and 250 µg/mL xanthine were added for J[µ1]L and N[µ1] (Sitia et al, 1990).

### E. coli cultivation

For plasmid amplification, *E. coli* XL1-blue were cultivated in LB medium supplemented with kanamycin (35 µg/mL) at 37 °C overnight. Protein expression was performed with *E. coli* BL21(DE3). The cells were cultivated in LB medium supplemented with kanamycin (35 µg/mL) at 37 °C. Expression was induced with 1 mM IPTG at an $OD_{600} = 0.6$–0.8. The cells were harvested after overnight expression at 37 °C.

### Cµ4tp domain and joining chain constructs

The genes of the murine Cµ4tp domain (UniProt: P01872, IGHM_MOUSE, Secreted isoform, E446–Y576, M added at position 1) and murine Joining Chain wild type (UniProt: IGJ_MOUSE, P01592, G22-D159, M added at position 1) as well as the mutants J Chain C13S-C15S and J Chain C69S-C72S were synthetized and cloned into the pET28b expression vector at the restriction sites NcoI and HindIII by the Thermo Fisher Scientific GeneArt Gene and Protein Synthesis Service. For further mutations of the J Chain gene, primers were designed using the NEBase-Changer tool and purchased from Eurofins Genomics GmbH. Mutations were generated by Q5 Site-Directed Mutagenesis Kinase, Ligase & DpnI (KLD) treatment. For the mutation, C13S C69S of

the J chain gene, the template pET28b_mJC_C13S, and primers for C69S were used. The final products were transformed into chemically competent *E. coli* XL1-Blue. Plasmids were isolated using the PureYield Plasmid Miniprep System and sequenced by Eurofins genomics TubSeq Service Supreme.

### Protein Purification from E. coli

The proteins were expressed as inclusion bodies and solubilized in 50 mM Tris (pH 8.5), 8 M Urea, 1 mM EDTA, and 14.3 mM β-mercaptoethanol at 4 °C while stirring overnight. Cell debris was removed by centrifugation at $46,000 \times g$ for 40 min at 4 °C. The supernatant was run over an XK Q-Sepharose FF column equilibrated with 50 mM Tris (pH 8.5), 1 mM EDTA, 5 M urea, and 4 mM DTT. Elution of the protein was performed with 50 mM Tris (pH 8.5), 1 mM EDTA, 5 M urea, 1 M NaCl, and 4 mM DTT. Fractions containing the protein were collected and applied to a Superdex 200 26/60 size exclusion chromatography (SEC) column previously equilibrated in PBS buffer (pH 7.4, 10 mM Na$_2$HPO$_4$, 1.8 mM KH$_2$PO$_4$, 2.8 mM KCl, 137 mM NaCl, 5 M Urea, and 4 mM DTT). For refolding and oxidation of the J chain, the protein was diluted to 0.1 mg/mL and dialyzed against PBS containing 1 M urea, 0.75 mM GSH, and 1 mM GSSG overnight at room temperature. Refolding and oxidation of the Cµ4tp domain was performed at a protein concentration of 0.1 mg/mL at 4 °C in three dialysis steps: (1) partial refolding in PBS with 1 M urea for 4 h, (2) oxidation in PBS with 1 M urea, 1 mM GSH, and 4 mM GSSG overnight, and (3) dialysis against PBS with 1 mM DTT for 4 h at room temperature to reduce the dimers into monomers. In a final purification step, the proteins were concentrated and applied to a Superdex 75 10/300 size exclusion chromatography (SEC) column equilibrated with PBS buffer or PBS buffer with 1 mM DTT in the case of Cµ4tp. Fractions containing predominantly monomeric J chain or Cµ4tp domain were pooled. The mass of the purified proteins was confirmed using matrix-assisted, laser desorption ionization, time-of-flight mass spectrometry.

### CD-spectrometry

The secondary structure of refolded and oxidized recombinant JC WT was analyzed by CD spectroscopy. A Chirascan-plus (Applied Photophysics Limited, UK) was used for the measurement. The proteins were measured in a 1.0 mm cuvette (Hellma), in PBS pH 7.4, at a concentration of 10 µM and a temperature of 25 °C. Data were analyzed and plotted using R 3.3.0+ and RStudio (2023.12.1 + 402) (R Core Team (2023)).

### Analytical and preparative SE-HPLC

Analytical SE-HPLC was used to assay the oligomeric state of Cµ4tp in the presence of the J chain. Samples contained 1 mg/mL Cµ4tp, 0.5 mg/mL (if not stated otherwise) of J chain, and protease inhibitor (SERVA Mix HP). Prior to preparing the sample, DTT was removed from Cµ4tp to ensure an oxidizing milieu. For all measurements, a Superdex 200 increase 10/300 column (GE Healthcare) connected to a Shimadzu HPLC system, operated by LC Solution 1.3.2 was used. Samples were kept at 25 °C, and the column was run in PBS buffer (pH 7.4, 10 mM Na$_2$HPO$_4$, 1.8 mM KH$_2$PO$_4$, 2.8 mM KCl, 137 mM NaCl) at a flow rate of 0.5 mL/min at room temperature. For each measurement, a sample volume of 10 µL was injected, and absorbance at 230 nm was detected. Preparative SE-HPLC was run at the same conditions with 500 µL of sample and was performed to collect

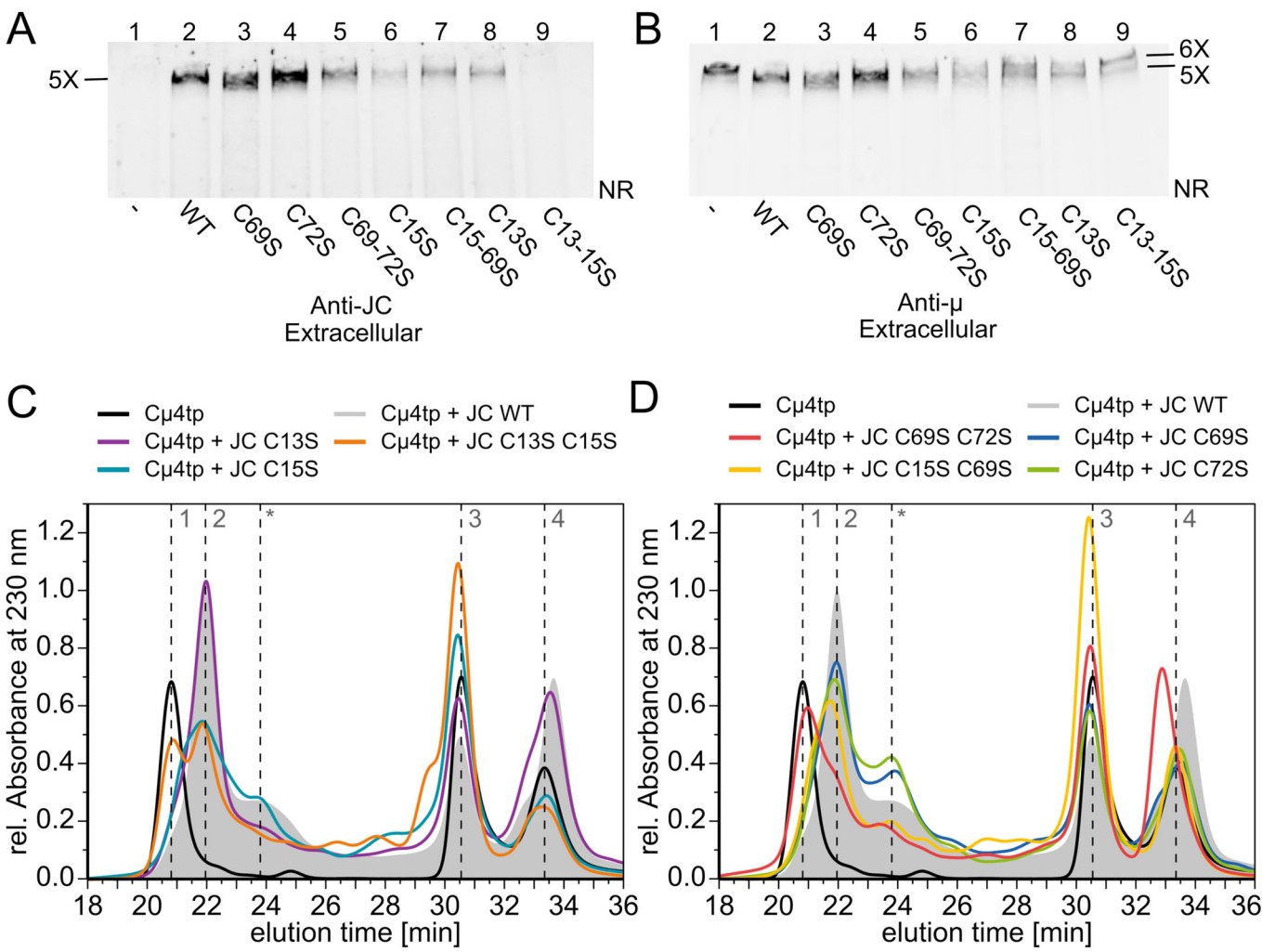

**Figure 7. Disulfide rearrangements during polymerization-driven JC folding.**

(A, B) Role of the JC cysteines in the assembly of secretion-competent IgM pentamers. Aliquots of the supernatants of HEK293 transfectants co-expressing H-Cμ234tp and the indicated JC mutants (C69, C72, C69-72S, C15S, C15-69S, C13S, and C13-15S) were resolved under nonreducing conditions and blots sequentially decorated with anti-JC and anti-μ, as indicated. Only the top part of the gels is shown here. Note that the C13-15S mutant is very poorly inserted in pentameric IgM and prevents only partly the secretion of hexamers. The experiments were repeated twice. (C, D) Aberrant interactions of JC lacking key cysteines with Cμ4tp in vitro. JC mutants C13S, C15S, C13S-C15S (panel C) and C69S, C72S, C15S-C69S, C69S-C72S (panel D) were incubated with Cμ4tp for 24 h at room temperature and analyzed by SE-HPLC. Their profiles are compared to WT JC, whose SEC pattern is shown as a gray background. Species: Peak 1: (Cμ4tp$_2$)$_6$. Peak 2: (Cμ4tp$_2$)$_5$-JC. The shoulder, indicated by an asterisk (*), contains uncharacterized assemblies of Cμ4-tp and JC. Peak 3:Cμ4tp$_2$ or JC. Peak 4: Cμ4tp. The experiments were performed in technical triplicates. Source data are available online for this figure.

fractions for SDS-PAGE analysis and limited proteolysis. Data were analyzed and plotted using R 3.3.0+ and RStudio (2023.12.1 + 402)(R Core Team (2023)).

### Limited proteolysis

Limited proteolysis was employed to test whether the different assembly species are protected from digestion. Cμ4tp pentamers with J chain were obtained by incubation of Cμ4tp (1 mg/mL) with J chain (0.5 mg/mL) for 24 h at room temperature and selectively isolated by separation with a Superdex 200 increase 10/300 column connected to a Shimadzu HPLC system equipped with a fraction collector. Hexamers were obtained by incubation of Cμ4tp (1 mg/mL). Cμ4tp monomers and JC were taken directly after purification. The species to be tested were diluted to 0.3 mg/mL with assay buffer (100 mM Tris/HCl pH

7.8, 100 mM NaCl, 10 mM CaCl$_2$), and the reaction was started by adding 1.2 μL proteinase K solution (0.2 mg/mL proteinase K in assay buffer, final substrate/enzyme ratio 1/250). At each time point, 20 μL aliquots were taken and mixed with 10 μL stop solution (6 mM PMSF, 6 μL 5x SDS-PAGE sample buffer: 300 mM Tris/HCl pH 6.8, 50% (v/v) glycerol, 10% (w/v) SDS, 0.05% (w/v) bromophenol blue) and kept on ice. The sample at 0 min was taken prior to the addition of proteinase K. For analysis, SDS-PAGE was performed using SERVAGel TG PRiME 14% gels (Serva) according to manufacturer's instructions.

### Cross-linking sample preparation

J-chain incubation/treatment.   About 15 μL of protein (≈1 mg/mL) were mixed with 15 μL of 8 M Urea in 50 mM Tris and incubated for 30 min at room temperature. As a control, one sample was reduced

with 1.5 μL of 100 mM TCEP, while 1.5 μL of water were added to the non-reduced native samples. All samples were further denatured at 56 °C for 30 min. Then 4.5 μL of 100 mM 2-Iodoacetamide were added and the samples were incubated for 20 min at room temperature in the dark. Samples were then precipitated with 40 volumes of ice-cold ethanol and incubated at –20 °C for 1 h. The samples were centrifuged at 16,000 rpm for 8 min and supernatant (s/n) was discarded. The pellets were then washed with 200 μL of ethanol, the s/n was discarded, and the samples were air-dried for 10 min. Then 100 μL of 50% acetonitrile (ACN) were added and the samples were dried in a SpeedVac concentrator for 2 h. The pellets were diluted in 49 μL of 2 M Urea, 10 mM CaCl₂, and 0.2 M Tris pH 6.7. Then, 1 μL of Trypsin (0.5 μg/μL) was added, and the reaction was held overnight at 37 °C. The samples were then additionally cleaved with 5 μL of 0.2 μg/μL Asp-N protease and incubated for 8 h at 37 °C. To stop the reaction, 1 μL of 100% formic acid (FA) was added, and the samples were stored overnight at −80 °C. The samples were then desalted with the pre-equilibrated in 70 μL of methanol and pre-washed with 70 μL of 0.5% FA double C18 layer stage tips. The samples were applied onto the tips and washed three times with 70 μL of 0.5% FA by mild centrifugation (400 × g). The peptides were eluted with two times 30 μL of 80% ACN and 0.5% FA via centrifugation and dried in a SpeedVac concentrator for 2 h. For the MS/MS measurements, the peptides were dissolved in 25 μL of 0.5% FA and sonicated in an ultrasonic bath at room temperature for 15 min. The peptides were filtered with 0.22 μM centrifugal filters for 1 min at 7000 × g.

Cross-linking - MS/MS measurement.   Samples were transferred to MS vials (Thermo Scientific, SureSTART) and analyzed via HPLC-MS/MS using the Vanquish Neo UHPLC (Thermo Fisher) equipped with a PepMap Neo 5 μm C18 300 μm × 5 mm Trap Cartridge (Thermo Fisher Scientific) and Aurora Ultimate separation columns (third generation, 20 cm nanoflow UHPLC compatible, ion optics) with a Nanospray Flex Ion Source coupled to Orbitrap Eclipse Tribrid instrument (Thermo Fisher). Vanquish Neo UHPLC was operated in Trap-and-Elute-Injection mode. Samples were loaded onto the trap column, and the subsequent separation was carried out with a flow rate of 400 nL/min using buffer A (0.1% FA in water) and buffer B (0.1% FA in acetonitrile). The separation column was heated to 40 °C. The analysis was started with a gradient from 5 to 28% buffer B for 30 min, a second gradient from 28% B to 40% B within 5 min, and a final increase to 90% B in 0.1 min. Isocratic washing with 90% B was performed for 9.9 min. For the wash and equilibration, the following settings at the Vanquish Neo system were applied: 5% buffer B. For the separation column, fast equilibration was enabled, and the equilibration factor was set to 3. For the trap column, fash wash and equilibration and zebra wash was enabled (Zebra Wash Cycles 2, equilibration factor set to automatic).

The Orbitrab Eclipse mass spectrometer was run in a data-dependent mode with a cycle time of 1 s. Internal real-time mass calibration was performed using user-defined lock mass (m/z = 445.12003, positive). In the orbitrap, full MS scans were collected in a scan range of 300–1500 m/z at a resolution of 120,000 and an AGC target of 3e6 with 80 ms maximum injection time. The most intense ions (charge states 2–12) were selected for MS2 scan with a minimum intensity threshold of 5.03e3 and isotope exclusion and dynamic exclusion (exclusion duration: 30 s) enabled. Peaks with unassigned charges or a charge of +1 were excluded. MS2 spectra were collected at a resolution of 60,000 with an AGC target set to standard and a maximum injection time of 100 ms. Isolation was conducted in the

quadrupole using a window of 1.6 m/z. Fragments were generated using higher-energy collision-induced dissociation (HCD, normalized collision energy: 30%) and finally detected in the Orbitrap. Data were acquired using Thermo Scientific Foundation software version 3.1sp9 and Xcalibur version 4.6. The experiments were performed in three independent replicates.

### Cross-linking MS data evaluation

The .raw files were converted to .mzXML with ProteoWizard converter (Chambers et al, 2012) and analyzed with Kojak (version 2.0.0 alpha 22 (Hoopmann et al, 2023, 2015)). The crosslinker was set to SS (disulfide bond) with a mass shift −2.016 Da for a crosslink and 0 Da for a monolink. Carbamidomethylation (57.02146) was set as fixed, and oxidation of methionine (15.994915) as variable modifications. The results were validated by percolator (version 2.08) (Käll et al, 2007) and visualized with ProXL (Riffle et al, 2016). The PSM q-value was set to <0.05. The figures include the crosslinks identified in all replicas. The mass spectrometry data were uploaded to the ProteomeXchange Consortium via PRIDE submission tool with identifier PXD048614.

### Cell transfection and secretion

HEK and ERp44^KOHeLa cells were transfected with jetPEI (Polyplus-transfection) following the protocol of the manufacturer. After transfection, cells were cultured for 48 h before biochemical analyses. For analyses of secreted IgM, cells were incubated with minimal essential medium (Opti-MEM) for 4 h. Cell culture SNs were then collected and treated with 10 mM N-ethylmaleimide (NEM; Sigma-Aldrich) to block disulfide interchange and protease inhibitors (Roche).

### Cell lysis and protein precipitation from supernatant

For Western blot assays, cells were detached and washed once in PBS with 10 mM NEM. Cells were then lysed in radioimmunoprecipitation assay (RIPA) buffer [150 mM NaCl, 1% 4-hydroxy-3-nitrophenyl acetyl (Nonidet P-40; Sigma-Aldrich), 0.1% SDS, 50 mM Tris/HCl (pH 8.0)] supplemented with 10 mM NEM and protease inhibitors (Roche) for 20 min on ice. Supernatants (SN) were concentrated by precipitation with 10% v/v TCA. Aliquots from lysates and SN were resolved on SDS/PAGE under reducing or nonreducing conditions. Proteins were then transferred to nitrocellulose and blotted with goat anti-mouse IgM (μ chain) Alexa Fluor 647 (Invitrogen Molecular Probes), goat anti-mouse L-HRP (Southern Biotech), rabbit anti-HaloTag (Promega) antibody, monoclonal anti-ERp44 antibodies (clones 36C9) or with rabbit anti-JC antibody (Fagioli and Sitia, 2001). Signals were acquired by Typhoon FLA 9000 (Fujifilm) and Chemidoc Imaging System (UVITEC).

### H-Cμ234tp plasmid production

The gene for H-Cμ234tp with a signal peptide was purchased by Eurofins Genomics and cloned into pcDNA3 vector via HindlII and Xbal restriction sites. The mutation C575A was introduced using the following primers purchased by Metabion: GGCGGCACCG CATATTGAAAG, CTTTCAATATGCGGTGCCGCC.

### Radioactive pulse and chase

Cells were incubated for 30 min in DMEM without methionine and cysteine supplemented with 1% dialyzed FCS, pulsed with 35S-labeled aminoacids (Easy Tag, Perkin Elmer), washed and chased in

complete medium for the indicated times. After different times, cells were treated with 10 mM NEM and lysed in RIPA as described (Mossuto et al, 2014). Immunoprecipitates were resolved on SDS–PAGE under reducing and nonreducing conditions, and gels were dried and visualized by autoradiography.

### HALO time-course

HEK293T cells were co-transfected with H-Cμ234tp and JC-myc or H-Cμ234tp and empty plasmid. The day after transfection, cells were split. Forty-eight hours after transfection, cells were incubated with minimal essential medium (Opti-MEM) and 100 nM of green HALOtag fluorescent ligand (R110Direct, Promega) for 75 min. Then, cells were gently washed-out with Opti-MEM (twice) and kept with 100 nM of red HALOtag fluorescent ligand (TMRDirect, Promega) for different incubation times. Cells were then lysed with RIPA adding 10 mM *N*-ethylmaleimide (NEM; Sigma-Aldrich) to block disulfide interchange and protease inhibitors (Roche).

## Data availability

The mass spectroscopic datasets produced in this study are available in the following databases: mass spectrometry data has been deposited to ProteomeXchange Consortium via the PRIDE repository with the project identifier PXD048614, "Disulfide mapping of J-chain by cross-linking MS" (https://www.ebi.ac.uk/pride/archive/projects/PXD048614).

The source data of this paper are collected in the following database record: biostudies:S-SCDT-10_1038-S44318-024-00317-9.

## Peer review information

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

## Acknowledgements

We thank Tiziana Anelli, Antonino Cattaneo, Marco Dalla Torre, Matthias Feige, and Michael Parkhouse for helpful discussions, Claudio Fagioli and Roberta Mancini for technical help with radioactive pulse chases, and Michaela Fiedler, Alexandra Geißler, and Katja Bäuml for MS measurements and method optimization. This work is dedicated to the memory of Michael Neuberger and Cèsar and Celia Milstein, pioneers, mentors and friends. This work was supported in part through grants from Associazione Italiana per la Ricerca sul Cancro (IG 23235), Ministero Università e Ricerca (PRIN XA5J5N) to RS, and Deutsche Forschungsgemeinschaft (DFG 5031251) to JB.

## Author contributions

**Chiara Giannone**: Conceptualization; Formal analysis; Validation; Investigation; Methodology; Writing—original draft; Project administration; Writing—review and editing. **Xenia Mess**: Data curation; Formal analysis; Investigation; Methodology; Writing—original draft; Writing—review and editing. **Ruiming He**: Formal analysis; Validation; Methodology. **Maria Rita Chelazzi**: Formal analysis; Validation; Investigation; Methodology. **Annika Mayer**: Validation; Investigation; Methodology. **Anush Bakunts**: Formal analysis; Validation; Investigation; Methodology. **Tuan Nguyen**: Formal analysis; Validation; Investigation; Methodology. **Yevheniia Bushman**: Investigation; Methodology. **Andrea Orsi**: Formal analysis; Supervision; Validation; Investigation; Methodology. **Benedikt Gansen**: Validation; Investigation. **Massimo Degano**: Conceptualization; Formal analysis; Validation; Investigation; Writing—original draft. **Johannes Buchner**: Conceptualization; Resources; Supervision; Funding acquisition; Validation; Writing—original draft; Project administration; Writing—review and editing. **Roberto Sitia**: Conceptualization; Supervision; Funding acquisition; Validation; Writing—original draft; Project administration; Writing—review and editing.

Source data underlying figure panels in this paper may have individual authorship assigned. Where available, figure panel/source data authorship is listed in the following database record: biostudies:S-SCDT-10_1038-S44318-024-00317-9.

## Funding

## Disclosure and competing interests statement

The authors declare no competing interests.

# Expanded View Figures

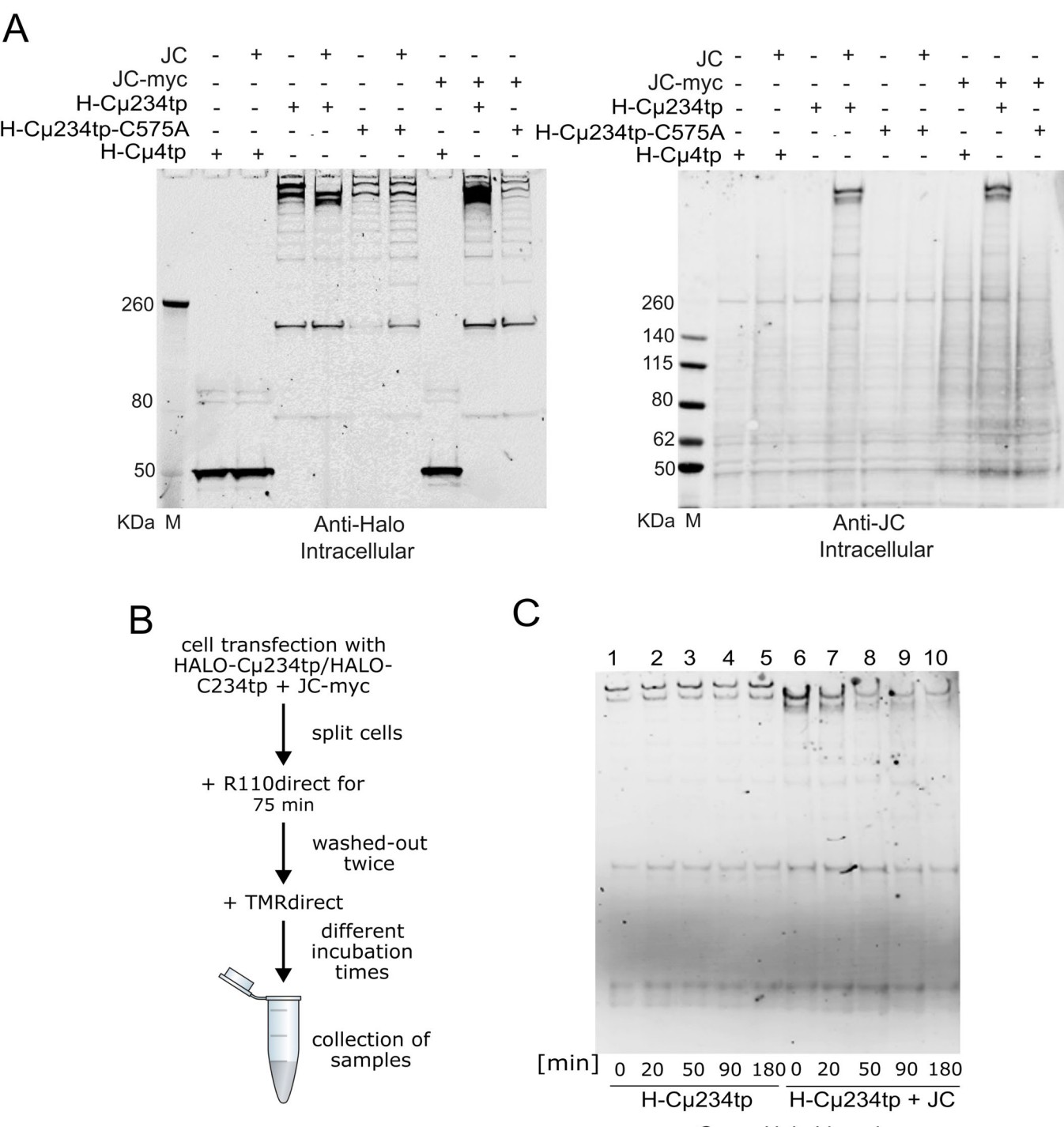

**Figure EV1. JC outcompetes the sixth μ2L2 subunit in associating with nascent pentamers.**

(A) HEK293 cells expressing H-Cμ234tp chains with a cysteine (WT) or alanine in the penultimate position (C575A) were co-transfected with myc-tagged JC or empty plasmid, as indicated. Aliquots of their lysates were resolved under nonreducing conditions and blots were decorated with Halo ligands (left panel) or anti-JC (right panel). (B). Halo time-course and kinetics of IgM assembly. As schematically summarized, HEK cells were co-transfected with H-Cμ234tp with or without JC-myc. Forty-eight hours after transfection, cells were incubated with green HALOtag fluorescent ligand (R110Direct, Promega) for 75 min, to label all preexisting molecules. Then cells were washed and incubated with red HALOtag fluorescent ligand (TMRDirect, Promega) for the indicated times (minutes). Cell lysates were analyzed electrophoretically (panel (C)) under nonreducing conditions, and the gel was directly analyzed for fluorescent R110 ligand. Note that hexamers are no longer detectable upon JC addition. JC do not favor the accumulation of dimers of H-Cμ234tp dimers, consistent with their preferential binding to nascent pentamers. As expected, H-Cμ234tp assembly species labeled before the addition of the red label tend to disappear with time.

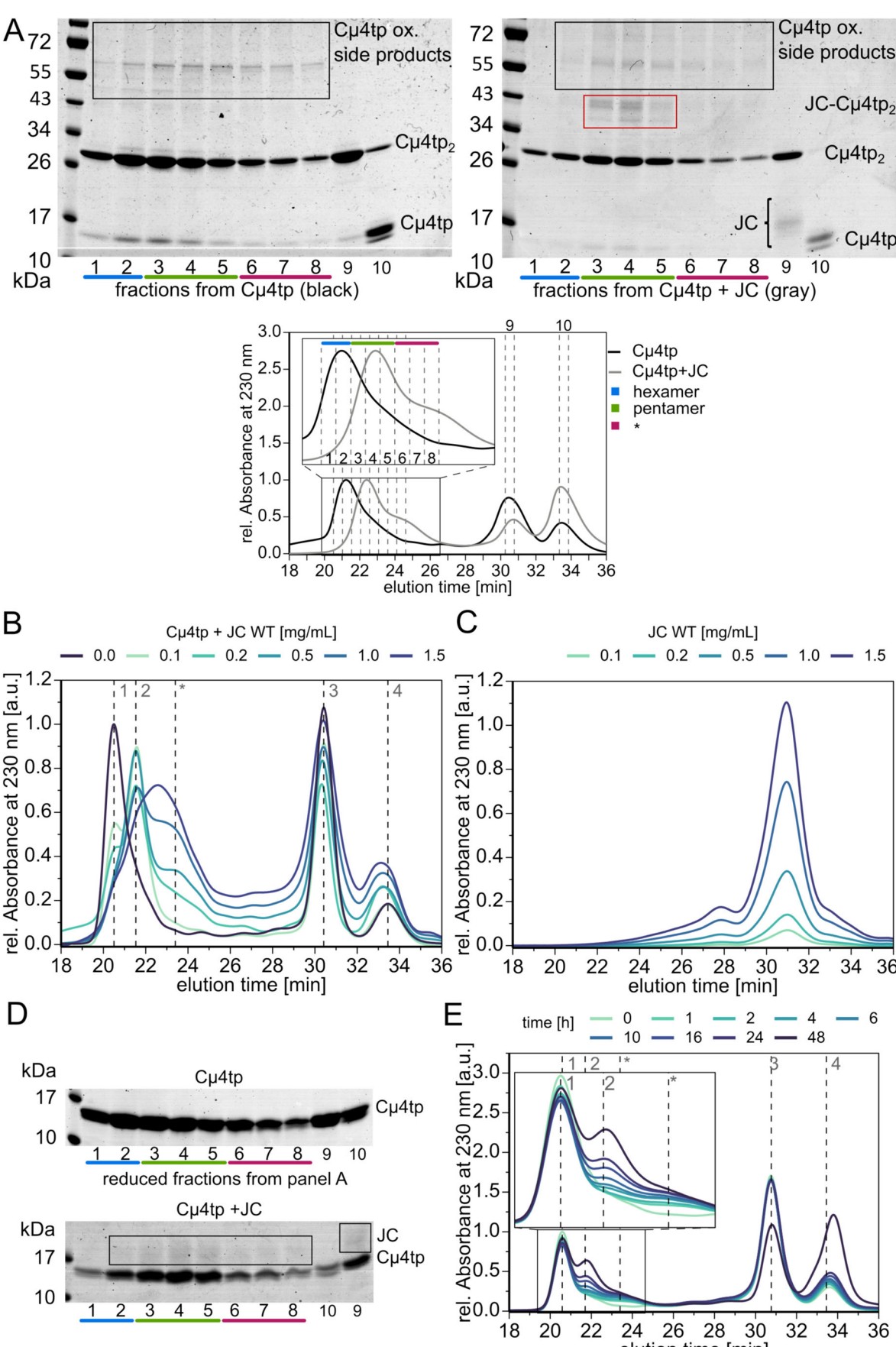

◀ **Figure EV2. JC outcompete Cμ4tp₂ subunits to complete stable hexamers.**

(A) Composition of the fractions obtained from preparative HPLC-SE. Aliquots of the high molecular peaks obtained by incubating Cμ4tp (1 mg/mL) with (black line) or without JC (0.5 mg/mL) (gray line) were applied to SE-HPLC and 250 μL and the indicated fractions (lower panel) analyzed by SDS-PAGE under nonreducing conditions (top two panels) or protease sensitivity assays (panel **D**). Fractions 1–2 (blue), correspond to hexamers, fractions 3–5 (green) to pentamer, fractions 6-8 (red) to the shoulder peak *, whilst fractions 9 and 10 contain JC+Cμ4tp dimers, or mainly Cμ4 monomers, respectively. As expected, species overlap to some degree. In the presence of JC, additional bands corresponding to a Cμ4tp₂-JC species are visible on SDS-PAGE (left panel, red box) in the pentamer and to a lesser extent in the * fractions. These bands are absent when Cμ4tp incubated alone. Cμ4tp on its own forms hexamers, which disassemble to dimers on SDS-PAGE (right panel). Some Cμ4tp oxidation side product bands (both panels, black boxes) are evident in the upper parts of both gels. These side products might contribute to the formation of the * shoulder. (B, C) JC outcompetes the sixth Cμ4tp₂ subunit in vitro. Increasing the amount of JC (from 0 to 1.5 mg/ml) added to 1 mg/ml Cμ4, progressively inhibits the formation of (Cμ4tp₂)₆ "hexamers" in favor of (Cμ4tp₂)₅-JC "pentamers" (panel **B**). At their highest concentration, a broad shoulder appears that presumably corresponds to aberrant assemblies in various stoichiometries. In the absence of Cμ4tp (panel **C**), most JC accumulates as a broad peak. A higher molecular weight shoulder was also observed, indicative of multimeric JC species, consistent with the smeary appearance of JC in cells (see Figs. 5 and EV3, panels **A, B**). (D) Reducing SDS-PAGE of the fractions shown in panel A confirms that JC is covalently linked to Cμ4tp in the oligomeric assemblies. All chromatograms were recorded thrice. (E) JC cannot be inserted in preformed (Cμ4tp₂)₆ hexamers. Preformed (Cμ4tp₂)₆ hexamers were incubated with 0.5 mg/mL amounts of JC for 48 h at room temperature and finally analyzed by SEC. The size of peak 1 remains constant, indicating that once formed, hexamers are rather stable. The increase in Peak 2 reflects the formation of pentamers during the incubation.

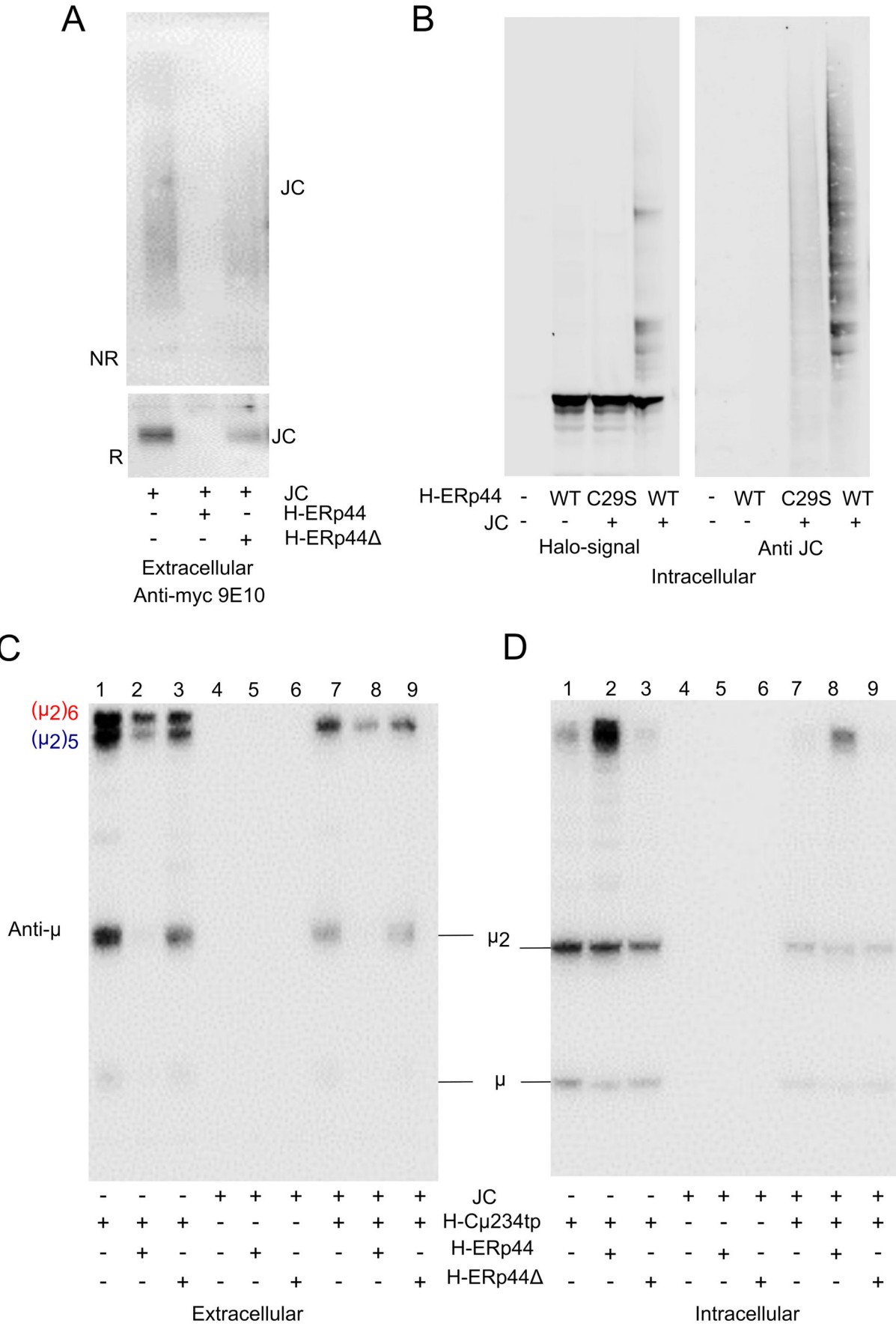

◀  **Figure EV3.  ERp44^KO cells secrete JC rich in heterogeneous, non-native disulfides.**

(A) When aliquots from the supernatants of ERp44^KO transfectants expressing myc-tagged JC are resolved electrophoretically under nonreducing conditions, anti-myc antibodies decorate a continuous smear. In contrast, DTT reduces the smear, and JC-myc chains accumulate as a 20–25 kDa band. Without alkylation, some intra-chain disulfide bonds may form during electrophoresis. Expression of WT ERp44 (center lane), but not of its ΔRDEL mutant (right lane), inhibits JC-myc secretion. All chromatograms were recorded thrice. (B) ERp44 binds JC via its C29. HeLa-ERp44^KO cells were transfected as indicated, and their lysates were run under nonreducing conditions and blotted with Halo ligand (left) or anti-JC. H-ERp44 forms many abundant complexes with JC that are not formed by a mutant lacking C29. Not all bands detected by anti-JC contain H-ERp44 (right panel). These data confirm that ERp44 efficiently retains JC through C29-dependent reversible disulfide bonds. (C, D) ERp44 prevents the secretion of unpolymerized IgM. HeLa-ERp44^KO cells were transfected as indicated, and aliquots from their supernatants (C) or lysates (D) resolved under nonreducing conditions and decorated with anti-μ antibodies. ERp44^KO HeLa cells secrete abundant H-Cμ234tp$_2$ subunits (D, lane 1). Co-expression of wild-type ERp44 restores retention of these incomplete subunits and promotes polymerization (panel D, compare lanes 1 and 2). H-ERp44 expression boosts also the formation of JC-containing pentamers (panel D, lanes 7-8), dampening the secretion of polymers, irrespective of the presence of JC (panel C lanes 7-8). H-ERp44Δ lacks the RDEL motif.

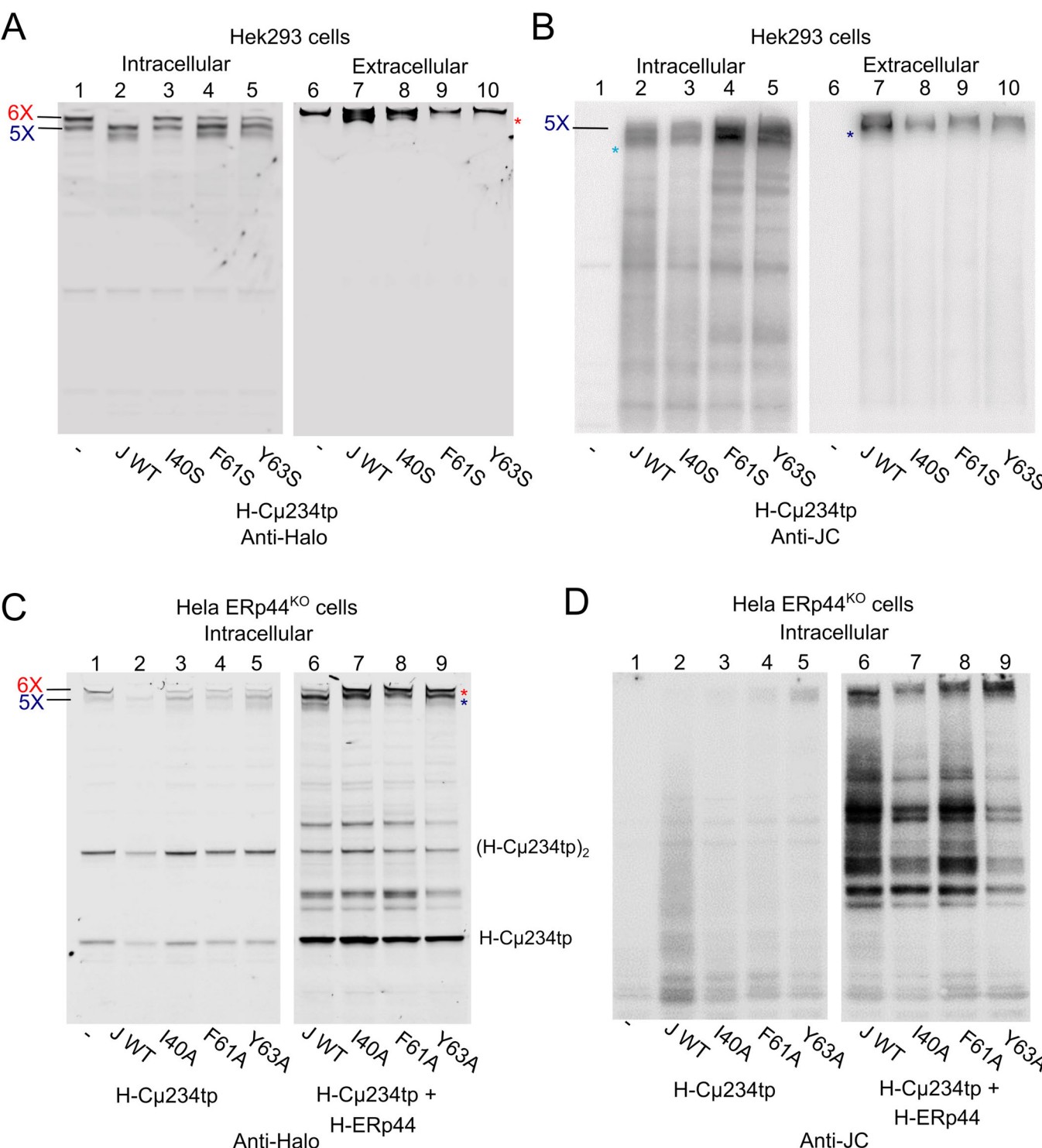

◀ **Figure EV4. Conserved hydrophobic residues drive assembly of secretion-competent IgM pentamers.**

(A, B) Essential roles of hydrophobic interactions. Replacing I40, F61, and Y63 with serine yields phenotypes similar or slightly more evident than alanine substitutions. Note that the three mutants are poorly inserted into secretion-competent pentamers (dark blue asterisk). As a result, hexamers (red asterisk) remain the most abundant intracellular species recognized by Halo ligands, and the sole one to be abundantly secreted by cells expressing ERp44. An uncharacterized form is retained intracellularly (light blue asterisk). Anti-JC antibodies intensely decorate F61S and Y63S in polymers accumulating intracellularly (lanes 4-5 panel **B**), but these are poorly secreted (lanes 9-10). Consequently, virtually only hexamers are detected by Halo ligands in the supernatants (lanes 9-10 panel **A**). (C, D). ERp44 binds JC as well as unpolymerized IgM intermediates. The lysates of ERp44^KO cells expressing H-Cμ234tp, rescued with or without WT H-ERp44, were decorated with Halo ligands or anti-JC. As expected, overexpression of ERp44 prevents JC and H-Cμ234tp secretion, increasing their intracellular accumulation. The abundance of anti-JC reactive covalent complexes in lanes 6–9 (panel **D**) indicates that ERp44 has a high affinity for JC.

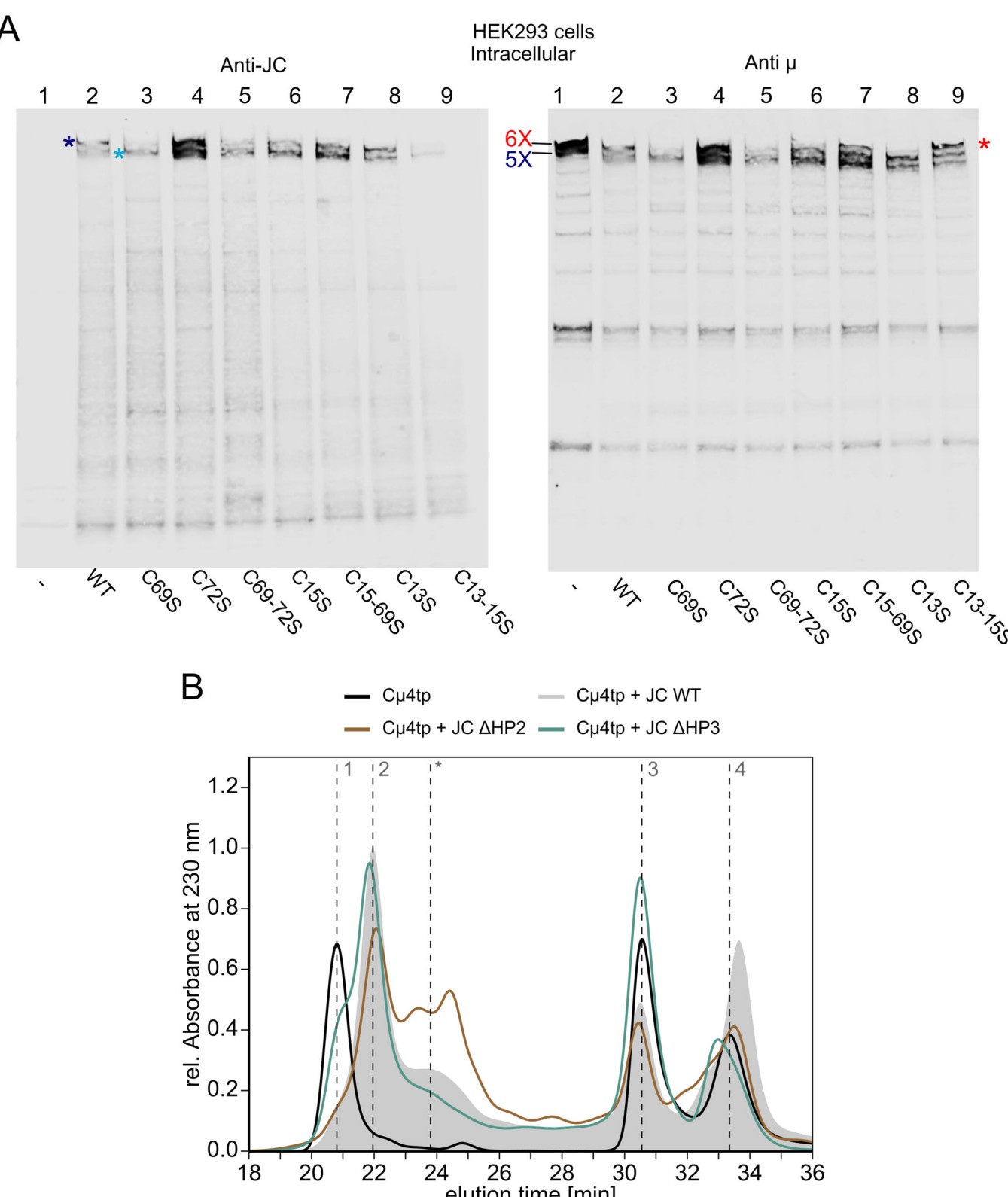

◀ **Figure EV5. Reactivity of the JC cysteine mutants in cellula and in vitro.**

(A) Pivotal role of C13 and C15. The lysates of HEK293 transfectants co-expressing H-Cμ234tp and the indicated JC mutants were resolved under nonreducing conditions and blots sequentially decorated with anti-JC and anti-μ, as indicated. Notably, the C13-15S mutant fails to bind IgM (lane 9) and forms mainly hexamers (red asterisk) detected with anti- μ, whilst the other mutants do bind IgM (dark blue asterisk) although with different efficiencies. An uncharacterized form is retained intracellularly (light blue asterisk). (B) The C-terminal part of JC is not essential for pentamer binding. SEC profiles on a Superdex 200 increase 10/300 GL in PBS. Cμ4tp (1 mg/mL) was incubated in vitro with WT or mutant JC (0.5 mg/mL) for 24 h at RT. Whilst the deletion of hairpin 3 and its flanking cysteines C110 and C135 (ΔHP3, turquoise) does not alter the SEC elution pattern (compare the blue trace with the gray pattern corresponding to WT JC), removal of hairpin 2 and the cysteines C72 and C92 (ΔHP2, brown trace) severely inhibits the formation of pentamers in vitro. These findings show that C110 and C135 are not vital for the formation of IgM pentamers. All chromatograms were recorded thrice.

