## [Peer Review File · The EMBO Journal]

How J-chain ensures the assembly of immunoglobulin IgM pentamers

Chiara Giannone, Xenia Mess, Ruiming He, Maria Rita Chelazzi, Annika Mayer, Anush Bakunts, Tuan Nguyen, Yevheniia Bushman, Andrea Orsi, Benedikt Gansen, Massimo Degano, Johannes Buchner, and Roberto Sitia

Corresponding author(s): Roberto Sitia (sitia.roberto@hsr.it), Johannes Buchner (johannes.buchner@tum.de), Chiara Giannone (giannone.chiara@hsr.it)

Review Timeline:

Submission Date:	15th Apr 24
Editorial Decision:	6th Jun 24
Revision Received:	19th Jul 24
Editorial Decision:	11th Sep 24
Revision Received:	23rd Oct 24
Accepted:	28th Oct 24

Editor: Ioannis Papaioannou

Transaction Report:

Dear Roberto,

Thank you again for submitting your manuscript EMBOJ-2024-117603 for consideration by The EMBO Journal, and for your patience during peer review. It has now been seen by two experts in the field, and we have received their comments, which you can find included below.

As you will see, both referees recognize the high quality of the presented data and the novelty of the findings elucidating further the mechanisms of IgM biogenesis. Referee #1 is also very supportive regarding the general relevance of the study for the fields of IgM assembly and immune function, and also finds it of broader interest to those working in the fields of protein folding, assembly, and quality control within the endoplasmic reticulum. The second referee, on the other hand, expresses doubts regarding the general significance of the results, as they find the conceptual advance incremental.

On balance, our decision is that we can offer publication of the manuscript in The EMBO Journal, provided that a number of technical and other concerns the referees raised will be sufficiently addressed in a revised version of the manuscript. I would like to invite you to submit this along with a detailed point-by-point response addressing all referees' comments. I should add that it is EMBO Journal policy to allow only a single round of major revision, and acceptance of your manuscript will therefore depend on the completeness of your responses in this revised version. Please let me know if you have any questions or comments that you would like to discuss with me.

We generally allow three months as standard revision time (September 5, 2024). As a matter of policy, competing manuscripts published during this period will not negatively impact our assessment of the conceptual advance presented by your study. However, we request that you contact us as soon as possible upon publication of any related work, to discuss how to proceed. Should you foresee a problem in meeting this three-month deadline, please let us know in advance and we may be able to grant an extension.

Thank you for the opportunity to consider your work for publication in The EMBO Journal. I look forward to your revision.

Best regards,

Ioannis

Instructions for preparing your revised manuscript

1. When you are ready to submit the revision, please upload:

- A Word file of the manuscript text (including legends of main Figures, EV Figures and Tables). Please make sure that changes are highlighted (or "tracked") to be clearly visible.

- Individual production-quality figure files (one file per figure). When assembling your figures, please refer to our figure preparation guidelines in order to ensure proper formatting and readability in print as well as on screen:

If the data shown in a figure are obtained from n {less than or equal to} 2, please use scatter plots showing the individual data points.

i. the name of the statistical test used to generate error bars and P values

ii. the number (n) of independent experiments (please specify technical or biological replicates) underlying each data point

(discussion of statistical methodology can be reported in the Materials and Methods section, but figure legends should contain a basic description of n , P , and the test applied)

iii. the nature of the bars and error bars (s.d., s.e.m.).

- A point-by-point response to the referees' comments, with a detailed description of the changes made (as a word file). All

referees' concerns must be fully addressed and their suggestions taken on board. When preparing your letter of response to the referees' comments, please bear in mind that this will form part of the Review Process File and will therefore be available online to the community. Please note that you have the possibility to opt out of the transparent process at any stage prior to publication by letting the editorial office know (contact@embojournal.org); if you do opt out, the Review Process File link will point to the following statement: "No Review Process File is available with this article, as the authors have chosen not to make the review process public in this case.". For more details on our Transparent Editorial Process, please visit our website: <https://www.embopress.org/page/journal/14602075/authorguide#transparentprocess>

- Expanded View (EV) files (replacing Supplementary Information) that are collapsible/expandable online. A maximum of 5 EV Figures can be typeset. EV Figures should be cited as "Figure EV1, Figure EV2" etc. in the text, and their respective legends should be included in the manuscript file after the legends of regular figures. See detailed instructions regarding Expanded View files here:

- For the figures that you do NOT wish to display as Expanded View figures, they should be bundled together with their legends in a single PDF file called "Appendix", which should start with a short Table of Contents (including page numbers). Appendix figures should be referred to in the main text as: "Appendix Figure S1, Appendix Figure S2" etc. Please see detailed instructions here: <https://www.embopress.org/page/journal/14602075/authorguide#expandedview>

- A complete author checklist, which you can download from our author guidelines (<https://www.embopress.org/page/journal/14602075/authorguide>). Please note that the checklist will also be part of the Review Process File.

2. Please note that no statistics should be calculated and shown in Figures if $n=2$. Please also note that each p value should be reported as an exact value.

3. Before submitting your revision, primary datasets (and computer code, where appropriate) produced in this study need to be deposited in appropriate public databases (see <https://www.embopress.org/page/journal/14602075/authorguide#dataavailability>).

In particular, you are kindly requested to deposit all mass spectrometry data produced in your study. The accession numbers and databases should be listed in a formal "Data availability" section (placed after Materials and Methods) that follows the model below (see also <https://www.embopress.org/page/journal/14602075/authorguide#dataavailability>):

Data availability

- RNA-seq data: Gene Expression Omnibus GSE46843 (<https://www.ncbi.nlm.nih.gov/geo/query/acc.cgi?acc=GSE46843>)
- [data type]: [name of the resource] [accession number/identifier/doi] ([URL or identifiers.org/DATABASE:ACCESSION])

*** All links should resolve to a page where the data can be accessed. ***

*** Please remember to provide in the Data availability section of your revised manuscript reviewer passwords if the datasets are not yet public. ***

*** The Data Availability Section is restricted to new primary data that are part of this study. In case you have no data that require deposition in a public database, please state so instead of referring to the database: "Our study includes no data deposited in public repositories." under the heading "Data availability". ***

4. Please check that the title and the abstract of the manuscript are brief, yet explicit, even to non-specialists. The length of the title should not exceed 100 characters, and the abstract should be a single paragraph not exceeding 175 words.

5. Please also note our reference format: <https://www.embopress.org/page/journal/14602075/authorguide#referencesformat>.

7. Please remember: digital image enhancement is acceptable practice, as long as it accurately represents the original data and conforms to community standards. If a figure has been subjected to significant electronic manipulation, this must be noted in the figure legend or in the "Materials and Methods" section. The editors reserve the right to request original versions of figures and the original images that were used to assemble the figure.

8. Our journal encourages inclusion of data citations in the reference list to directly cite datasets that were obtained from public

databases. Data citations in the article text are distinct from normal bibliographical citations and should directly link to the database records from which the data can be accessed. In the main text, data citations are formatted as follows: "Data ref: Smith et al, 2001" or "Data ref: NCBI Sequence Read Archive PRJNA342805, 2017". In the Reference list, data citations must be labeled with "[DATASET]". A data reference must provide the database name, accession number/identifiers, and a resolvable link to the landing page from which the data can be accessed at the end of the reference. Further instructions are available at: <https://www.embopress.org/page/journal/14602075/authorguide#referencesformat>.

9. We request authors to consider both actual and perceived competing interests. Please review our policy (<https://www.embopress.org/page/journal/14602075/authorguide#conflictsinterest>) and update your competing interests statement if necessary. Please name this section 'Disclosure and competing interests statement' and place it after the Acknowledgements section.

10. Please note that all corresponding authors are required to provide an ORCID ID upon submission of a revised manuscript (<https://orcid.org/>). Please find instructions on how to link your ORCID ID to your account in our manuscript tracking system in our Author guidelines (<https://www.embopress.org/page/journal/14602075/authorguide#authorshipguidelines>).

11. We use CRediT to specify the contributions of each author in the journal submission system. CRediT replaces the author contribution section, which should be removed from the manuscript. Please use the free text box to provide more detailed descriptions. See also guide to authors: <https://www.embopress.org/page/journal/14602075/authorguide#authorshipguidelines>.

13. We would also welcome the submission of cover suggestions or motifs to be used by our Graphics Illustrator in designing a cover.

14. Please use the link below to submit your revision:
<https://emboj.msubmit.net/cgi-bin/main.plex>

Referee #1:

The article entitled "How J Chain Ensures the Formation of IgM Pentamers" by Giannone et al. investigates the mechanism of IgM polymerization, focusing on how the joining chain (JC) is integrated into IgM pentamers. IgM antibodies, crucial for primary immune responses, can form either hexamers or pentamers, the latter including JC, which facilitates mucosal immunity through transcytosis. The study reveals that JC, initially an unstructured polypeptide with non-native disulfides, selectively binds to pentameric IgM by interacting with exposed hydrophobic β -sheets. This binding triggers JC folding and stabilizes the pentameric structure through disulfide rearrangements. The quality control protein ERp44 monitors IgM assembly, preventing the secretion of aberrant forms. This mechanism ensures the production of IgM with high avidity and appropriate stoichiometry for effective immune function.

The presented findings elucidate the previously unknown molecular mechanism behind the selective incorporation of JC into IgM pentamers, advancing our understanding of IgM biogenesis. Moreover, by clarifying how JC ensures the production of pentamers, these findings highlight the crucial role of JC in directing IgM towards mucosal immunity, enhancing our knowledge of immune system functioning. The study also provides insights into the broader principles of protein folding, assembly, and quality control within the endoplasmic reticulum, contributing to the fields of molecular biology and biochemistry.

Overall, this is very interesting research, and the data are important for understanding the basic principles of IgM biogenesis and immune function. However, I believe that the manuscript requires some improvement to better explain the conducted experiments.

Specific Comments

- While the scientific question is clear, the description of some experiments and reagents is not. For instance, in Fig. 1, it was not clear which constructs were transfected and which IgM components were cell endogenous. The authors state: "To decipher how JC is integrated into IgM polymers, we analyzed a panel of myeloma transfectants expressing JC, light (L), and secretory μ chains (Fig 1C): J[μ s] express all three chains and secrete NP-specific IgM polymers (lanes 1 and 5). N[μ 1] cells express μ s and JC (lanes 3 and 7), whilst J558L cells (lanes 2 and 6) express L and JC (Sitia et al., 1990; Fagioli & Sitia, 2001)." Are lanes 2 and 6 not transfectants? What are lanes 4 and 8, and why are they not mentioned in the results? The numbers of the lanes in Fig. 1C, particularly 5-8, are not aligned to the lanes.
- No controls for the transfections are shown.
- What are the high molecular weight bands in lanes 2 and 4 of Fig. 1C?

- The designations of the cell lysates are not intuitive. For instance: "J[μs] express all three chains and secrete NP-specific IgM polymers." Referring to this lysate as Jμλ would simplify reading and understanding.
- The authors stated, "The monomeric JC migrated as a diffuse band in non-reducing conditions (top panel). Consistent with a heterogeneous array of intra-chain disulfide bonds, JC collapsed into a sharp band upon DTT reduction (bottom panel)." Is this difference enough to draw this conclusion?
- No molecular weight marker is shown for the blots in Fig. 1 and 2.
- It is not clear why a myc-tagged JC was used as native JC could be detected with the anti-JC antibody used in Fig. 1.
- Unusual terms are used to describe the western blot techniques (e.g., "blots were sequentially hybridized with anti-J and anti-μ antibodies"). Hybridization is used for nucleic acids (RNA, DNA) and not for incubation of western blots with primary or secondary antibodies.

Minor Points

- Some figure legends contain information that belongs elsewhere. For instance, Fig. 1 legend states: "In JC containing pentamers, the only form for which structural information is available subunit 1 displays antiparallel tailpieces, whilst these run parallel to each other in subunits 2-5 (for further details see Fig EV5A)." However, no β-sheets for any μ2L2 subunit are shown in this figure.
- In the results text, the authors state: "JC caps the β-sandwich structure formed by the μtps in growing pentamers, adopting an arrangement that mimics subunit 1, with two antiparallel β-strands (β3 and β4) facing the μtps of subunits 4 and 5 (Fig EV5A)." However, in the figure legend to Fig EV5A, the authors state: "Note also how the two cyan beta sheets from the J chain are mimicking the μtps parallel arrangements in the pentamer core." This is confusing because it might imply that the J chain has parallel β-strands.

Referee #2:

IgM exists primarily in its pentameric form, comprised of five IgM monomers interconnected by the joining chain (JC). The study by Giannone et al. delved into the biochemical mechanisms governing the assembly of the IgM-JC complex. Their results suggest that the JC is the final player in completing the IgM pentamer, displacing the potential sixth IgM subunit. The ER chaperone ERp44 plays a crucial role in the assembly process by safeguarding against the secretion of defective JC conformers. This investigation offers a comprehensive insight into the intricacies of IgM pentamer formation. While the experiments were technically challenging, they were well conducted with precision. The significance of this study may be debatable as it does not seem to introduce fundamentally novel insights.

Suggestions for improvement:

1. Fig. 1C should include a molecular weight standard for reference. The description of "myeloma transfectants" needs to be elaborated in both the Results and Methods sections. Additionally, Fig. 1D lacks a loading control.
2. For Fig. 3, more data are needed to validate the quality of the Cμ4tp and J-chain proteins purified and refolded from *E. coli*. This could include SDS-PAGE and gel filtration analyses to demonstrate the purity of the "predominantly monomeric J chain or Cμ4tp domain." It is also essential to include a molecular weight standard for the gel filtration experiment, especially in Fig. 3A. While the *in vitro* reconstitution experiment is impressive, ensuring the data's credibility is crucial.
3. Fig. 4 should be moved to the supplemental figures for better organization.

Referee #1:

The article entitled "How J Chain Ensures the Formation of IgM Pentamers" by Giannone et al. investigates the mechanism of IgM polymerization, focusing on how the joining chain (JC) is integrated into IgM pentamers. IgM antibodies, crucial for primary immune responses, can form either hexamers or pentamers, the latter including JC, which facilitates mucosal immunity through transcytosis. The study reveals that JC, initially an unstructured polypeptide with non-native disulfides, selectively binds to pentameric IgM by interacting with exposed hydrophobic β -sheets. This binding triggers JC folding and stabilizes the pentameric structure through disulfide rearrangements. The quality control protein ERp44 monitors IgM assembly, preventing the secretion of aberrant forms. This mechanism ensures the production of IgM with high avidity and appropriate stoichiometry for effective immune function.

The presented findings elucidate the previously unknown molecular mechanism behind the selective incorporation of JC into IgM pentamers, advancing our understanding of IgM biogenesis. Moreover, by clarifying how JC ensures the production of pentamers, these findings highlight the crucial role of JC in directing IgM towards mucosal immunity, enhancing our knowledge of immune system functioning. The study also provides insights into the broader principles of protein folding, assembly, and quality control within the endoplasmic reticulum, contributing to the fields of molecular biology and biochemistry.

Overall, this is very interesting research, and the data are important for understanding the basic principles of IgM biogenesis and immune function. However, I believe that the manuscript requires some improvement to better explain the conducted experiments.

We thank the reviewer for their positive comments and constructive suggestions, which helped us to make the text clearer and easier to follow.

Specific Comments

1. While the scientific question is clear, the description of some experiments and reagents is not. For instance, in Fig. 1, it was not clear which constructs were transfected and which IgM components were cell endogenous. The authors state: "To decipher how JC is integrated into IgM polymers, we analyzed a panel of myeloma transfectants expressing JC, light (L), and secretory μ chains (Fig 1C): J[μ s] express all three chains and secrete NP-specific IgM polymers (lanes 1 and 5). N[μ 1] cells express μ s and JC (lanes 3 and 7), whilst J558L cells (lanes 2 and 6) express L and JC (Sitia et al., 1990; Fagioli & Sitia, 2001)." Are lanes 2 and 6 not transfectants? What are lanes 4 and 8, and why are they not mentioned in the results? The numbers of the lanes in Fig. 1C, particularly 5-8, are not aligned to the lanes.

We have now clarified the origin of the cell lines analyzed and added the information missing in figure 1C. We also aligned the labels to the lanes.

2. No controls for the transfections are shown.

The transfection controls are part of figure 1C. Lanes 1-2 show J558L (which expresses endogenous J and Lambda chains) while lanes 3-4 show NSO (which express only J). Lanes 1 and 3 show the corresponding transfectants expressing

NP-specific μ chains. As expected, anti- μ antibodies interact only with bands in lanes 1 and 3. The same blot had been incubated also with anti-lambda antibody, clearly confirming the identity of the cells and the specificity of the antibodies. The anti-lambda is now shown in the revised manuscript in Fig. 1C. The full image is attached below for the reviewers' eyes. Specificity controls of our anti-JC reagents are shown in figures 2A-B, 5A, 6C-E, 7A, EV1A and C; EV4B and D; EV5A. Additional serological data and extensive characterization of the cell lines and vectors used is available in our previous papers, cited in the present manuscript.

3. What are the high molecular weight bands in lanes 2 and 4 of Fig. 1C?

We share the referee's curiosity concerning the identity of the band that seems to appear in cells that do not express μ chains. With the available data, we can only hypothesize that the band represents a covalent complex containing one or more JC and/or other protein(s) of the early secretory pathway. In support of this hypothesis, JC can form mixed disulfides with many proteins when expressed in HeLa cells (Figure 5).

We added a sentence in the legend to the figure stating that the composition of the high molecular weight is unknown.

4. The designations of the cell lysates are not intuitive. For instance: "J[μ s] express all three chains and secrete NP-specific IgM polymers." Referring to this lysate as J μ λ would simplify reading and understanding.

We have changed the nomenclature and clarified the text and figure.

5. The authors stated, "The monomeric JC migrated as a diffuse band in non-reducing conditions (top panel). Consistent with a heterogeneous array of intra-chain disulfide bonds, JC collapsed into a sharp band upon DTT reduction (bottom panel)." Is this difference enough to draw this conclusion?

The reviewer addresses an important point. We think it is and the data shown later in the manuscript confirm this notion. Formal evidence that monomeric JC forms non-native disulfides is provided in Figure 4B, which describes the type and relative abundance of the disulfide bonds formed *in vitro* upon oxidation of purified JC.

Nonetheless, for the sake of precision, we modified the sentence as follows:

Interestingly, at all time-points tested; monomeric JC migrated with a more diffuse pattern in non-reducing conditions (top panel) than in reducing conditions (bottom panel), suggesting that JC may adopt multiple redox isoforms through the different arrangement of its intra-chain disulfide bridges, the nature of which is of particular interest in this study.

The reviewer may be interested by an intriguing point emerging from the results shown in Bertoli et al. 2004. In that study, the radioactive pulse chase experiments showed that newly made JC tends to form mainly intrachain bonds, whilst western blots show also abundant high molecular weight bands, implying inter-chain disulfides. This may explain some discrepancies (e.g. the migration of 'monomeric' J chains in panels C and D of figure 1).

6. No molecular weight marker is shown for the blots in Fig. 1 and 2.

We have added information on the markers used in Fig. 1, Fig. 2 and Fig. EV1.

7. It is not clear why a myc-tagged JC was used as native JC could be detected with the anti-JC antibody used in Fig. 1.

We thank the reviewer for this comment. The short answer is that the tag had been added to JC to provide an additional means to identify and immuno-purify the assembly intermediates and characterize unambiguously their composition. We added a revised figure showing that myc-tagged and untagged JC assemble similarly with H-C μ 234tp. Clearly, neither the capability of inserting into pentamers nor the electrophoretic patterns under non-reducing conditions are impaired by the presence of the C terminal tag. The entire gels are shown in EV 2A: here the last constant domain of IgM bound to Halo tag (H-C μ 4tp) was used to show that the only cysteine on the tailpiece was not sufficient in cell to form covalent oligomers that bind JC.

Figure 2A has been adjusted.

8. Unusual terms are used to describe the western blot techniques (e.g., "blots were sequentially hybridized with anti-J and anti- μ antibodies"). Hybridization is used for nucleic acids (RNA, DNA) and not for incubation of western blots with primary or secondary antibodies.

As suggested, we replaced "hybridize" with "decorate" or "incubate", in pages 4 and 26.

Minor Points

1. Some figure legends contain information that belongs elsewhere. For instance, Fig. 1 legend states: "In JC containing pentamers, the only form for which structural information is available subunit 1 displays antiparallel tailpieces, whilst these run parallel to each other in subunits 2-5 (for further details see Fig EV5A)." However, no β -sheets for any μ 2L2 subunit are shown in this figure.

We thank the referee for this suggestion. Accordingly, we edited the text and deleted the reference to parallel and antiparallel tailpieces from the legend to figure 1A.

2. In the results text, the authors state: "JC caps the β -sandwich structure formed by the μ tps in growing pentamers, adopting an arrangement that mimics subunit 1, with two antiparallel β -strands (β 3 and β 4) facing the μ tps of subunits 4 and 5 (Fig EV5A)." However, in the figure legend to Fig EV5A, the authors state: "Note also how the two cyan beta sheets from the J chain are mimicking the μ tps parallel arrangements in the pentamer core." This is confusing because it might imply that the J chain has parallel β -strands.

As suggested, we changed the text in the results and legend, referring to the previous Fig. EV5 (now Appendix Fig. S3 in the revised manuscript), and omitted the confusing reference to parallel and antiparallel tailpiece orientation.

Referee #2:

IgM exists primarily in its pentameric form, comprised of five IgM monomers interconnected by the joining chain (JC). The study by Giannone et al. delved into the biochemical mechanisms governing the assembly of the IgM-JC complex. Their results suggest that the JC is the final player in completing the IgM pentamer, displacing the potential sixth IgM subunit. The ER chaperone ERp44 plays a crucial role in the assembly process by safeguarding against the secretion of defective JC conformers. This investigation offers a comprehensive insight into the intricacies of IgM pentamer formation. While the experiments were technically challenging, they were well conducted with precision. The significance of this study may be debatable as it does not seem to introduce fundamentally novel insights.

We thank the reviewer for their kind words and constructive suggestions, which we have followed carefully. Perhaps not surprisingly, however, we respectfully disagree with the last sentence. The findings that JCs remain naturally unfolded, with many non-native disulfide bonds, until they encounter a nascent IgM pentamer, and yet can elude the stringent ER quality control mechanisms, open a series of relevant structural and evolutionary questions. *De gustibus non est disputandum...*

Suggestions for improvement:

1. Fig. 1C should include a molecular weight standard for reference.

We included a molecular weight (MW) standard for Figure 1C, However, we omitted it in the subsequent figures to reduce figure complexity.

The description of "myeloma transfectants" needs to be elaborated in both the Results and Methods sections.

We have simplified the text and added more information concerning the features of these transfectants. Additional information is available in previous papers of ours, cited in the present manuscript.

Additionally, Fig. 1D lacks a loading control.

This is a pulse chase experiment, and no loading control can be provided. Rigorous serological controls of the anti JC antibodies are shown in this (see Figs 1C and 2) and many previous papers of ours (see for instance Mancini et al., 2000; Fagioli and Sitia, 2001; Fagioli et al 2001).

2. For Fig. 3, more data are needed to validate the quality of the C μ 4tp and J-chain proteins purified and refolded from E. coli. This could include SDS-PAGE and gel filtration analyses to demonstrate the purity of the "predominantly monomeric J chain or C μ 4tp domain." It is also essential to include a molecular weight standard for the gel filtration experiment, especially in Fig. 3A. While the in vitro reconstitution experiment is impressive, ensuring the data's credibility is crucial.

We thank the referee for the positive words concerning the in vitro reconstitution. In response to the reviewer's query, we added a non-reducing 15% SDS gel demonstrating the purity of the JC mutants as well as the C μ 4tp domain (Appendix Fig. S2). As suggested, we added a molecular weight standard to Fig. 3A.

3. Fig. 4 should be moved to the supplemental figures for better organization.

We thank the reviewer for this suggestion. Although we agree that these are complex data to digest, we would like to keep it as a main figure, as it addresses a fundamental point of our story: that JCs are largely intrinsically disordered and rich in non-native bonds until they assemble with pentamers.

In conclusion, having answered to all the criticisms and comments of the reviewers, we are confident that you will find this revised manuscript suitable for publication in the EMBO J.

With our best regards

Roberto Sitia and Johannes Buchner

Dear Roberto, dear Johannes,

Thank you again for the submission of your revised manuscript to The EMBO Journal. Once again, I sincerely apologize for the rather slow review process on this occasion -which was due to the unavailability of the referees for a few weeks and editorial annual leave absences in our team- and thank you for your understanding and patience.

As I have already informed you, your revised manuscript has now been reassessed by referee #1 (their comments are included below), and I am glad to say that it is endorsed by the referee, who recognizes that the new version of the manuscript is significantly strengthened and improved. In light of this input and considering that all suggestions and concerns of both referees have been comprehensively addressed, we are happy to publish your study in The EMBO Journal.

From the editorial side, there are a few minor changes and corrections we need from you in a final version of your manuscript before we can proceed with its acceptance and publication:

- There is a co-author name discrepancy between the authors' list in the manuscript and our online manuscript handling system: "Ben Gansen" in the manuscript vs. "Benedikt Gansen" in our system. Could you please correct this so that the name appears identical in both the manuscript and the author's profile in the system?

- Your abbreviations list needs to be removed from the manuscript. Abbreviations should be defined in brackets after their first mention in the text, not in a separate list of abbreviations.

- Please reduce the number of your keywords after the Abstract to 5; you currently have 10, but no more than 5 keywords can be listed.

- Thank you for adding a Data availability statement with the access information for your mass spectrometry data. Please make sure that the datasets will be publicly available at the time of publication. The reviewer access information can now be removed from this statement. Please also include the specific URL link to your dataset in the PRIDE repository.

- The author contributions statement should be removed from the manuscript file. Instead, we use CRediT to specify the contributions of each author in the journal submission system. Please feel free to use the free text box to provide more detailed descriptions during submission. See also our guide to authors for more information: <https://www.embopress.org/page/journal/14602075/authorguide#authorshipguidelines>.

- Please include a separate conflict-of-interest statement with the heading "Disclosure and competing interests statement", following the Data availability statement.

- We noticed that callouts for Figure 5C are missing from the main text; please add at least one callout for this figure panel where appropriate.

- Please add the heading "Appendix" at the beginning of your single Appendix pdf file; the brief Table of Contents of this file should also be included on the first, title page of the Appendix.

- Thank you for uploading the requested source data. Could you please add a clarifying statement in the notes of your Source Data checklist (there is space for this at the bottom of each page) that the Source Data for Figure 4C have been deposited to an external repository (PRIDE)?

- Please note that EMBO press papers are accompanied online by:

A) a short (2 sentences) summary of the findings and their significance,

B) 2-5 short bullet points highlighting the key results, and

C) a synopsis image in .jpg or .png format that is exactly 550 pixels wide and 300-600 pixels high (the height is variable). Please note that the text needs to be legible at the final size.

Please upload this information along with your revised manuscript (the text for A and B should be provided in a separate Word file).

- Please change the heading "Materials and Methods" to "Methods".

- The Reagents and Tools table should only be uploaded as a separate file (choosing the file type "Reagents table"), not included in the main manuscript file. Please remove the instructions from the top and the examples from the bottom of your Reagents and Tools table.

- Please note that information related to "n" is missing in the legend of Figure 3b.

- Please note that the error bars should be defined in the legend of Figure 3b.
- Please note that the black and red asterisks are not defined in the legend of Figure 3e. This needs to be rectified.
- Please note that the black and blue asterisks are not defined in the legends of Figures EV 4b, EV 5a. This needs to be rectified.

Please also note that as part of the EMBO publications' Transparent Editorial Process, The EMBO Journal publishes online a Peer Review File along with each accepted manuscript. This File will be published in conjunction with your paper and will include the referee reports, your point-by-point response and all pertinent correspondence relating to the manuscript. You can opt out of this by letting the editorial office know (contact@embojournal.org). If you do opt out, the Peer Review File link will point to the following statement: "No Peer Review File is available with this article, as the authors have chosen not to make the review process public in this case."

We look forward to seeing a final version of your manuscript as soon as possible. Please let us know if you have any questions and use this link to submit your revision: <https://emboj.msubmit.net/cgi-bin/main.plex>

Best wishes,

Ioannis

Referee #1:

The authors have comprehensively addressed my previous concerns and suggestions. The additional experiments they performed significantly clarify and strengthen the findings. After these thorough revisions, I believe the manuscript is now much improved and acceptable for publication.

All editorial and formatting issues were resolved by the authors.

Dear Roberto, dear Johannes,

Congratulations on an excellent manuscript! I am very pleased to inform you that it has been accepted for publication in The EMBO Journal. Thank you very much for your comprehensive responses to the referee comments and the requested formatting changes.

If you have any questions, please do not hesitate to contact the Editorial Office. Thank you again for your contribution to The EMBO Journal. Working with you has been a pleasure!

Best wishes,

Ioannis
